

# The improvements to the regional South China Sea Operational Oceanography Forecasting System

Xueming Zhu[1,2], Ziqing Zu[1], Shihe Ren[1*], Yunfei Zhang[1], Miaoyin Zhang[1], Hui Wang[3,1*]

[1]National Marine Environmental Forecasting Center, Key Laboratory of Marine Hazards Forecasting, Ministry of Natural Resources, Beijing, 100081, China

[2]Southern Marine Science and Engineering Guangdong Laboratory (Zhuhai), Zhuhai, 519000, China

[3]Institute of Marine Science and Technology, Shandong University, Qingdao, Shandong, 266237, China

*Correspondence to*: Shihe Ren (rensh@nmefc.cn), Hui Wang(wangh@nmefc.cn)

**Abstract.** South China Sea Operational Oceanography Forecasting System (SCSOFS) had been built up and operated in National Marine Environmental Forecasting Center of China to provide daily updated hydrodynamic forecasting in SCS for the future 5 days since 2013. This paper presents comprehensive updates had been conducted to the configurations of the physical model and data assimilation scheme in order to improve SCSOFS forecasting skills in recent years. It highlights three of the most sensitive updates, sea surface atmospheric forcing method, tracers advection discrete scheme, and modification of data assimilation scheme. Scientific inter-comparison and accuracy assessment among five versions during the whole upgrading processes are performed by employing Global Ocean Data Assimilation Experiment OceanView Inter-comparison and Validation Task Team Class4 metrics. The results indicate that remarkable improvements have been achieved in SCSOFSv2 with respect to the original version SCSOFSv1. Domain averaged monthly mean root mean square errors decrease from 1.21 ℃ to 0.52 ℃ for sea surface temperature, from 21.6cm to 8.5cm for sea level anomaly, respectively.

## 1. Introduction

The South China Sea (SCS) is located between 2°30′S～23°30′N and 99°10′E～121°50′E, the largest in area and the deepest in depth, a semi-closed marginal sea in the western Pacific. Its area is about 3.5 million km², and its maximum depth is about 5300 m at the central region. It connects to the East China Sea by the Taiwan Strait to the northeast, to the North Pacific Ocean by the Luzon Strait (LUS) to the



east, to the Java Sea by the Karimata Strait to the south. Numerous islands, irregular and complex coastal boundaries, and drastic changes in bottom topography all together contribute to the great complex distribution of topography in the SCS.

The upper-layer basin-scale ocean circulations of the SCS are mainly controlled by the East Asian Monsoon (Hellerman and Rosenstein, 1983), showing a cyclonic gyre in winter and an anti-cyclonic gyre in summer (Mao et al., 1999;Chu and Li, 2000). The multi-scale oceanic circulation dynamical processes of the SCS are affected by various factors, i.e. the Kuroshio intrusion through the LUS (Nan et al., 2015;Farris and Wimbush, 1996;Liu et al., 2019), internal waves (Li et al., 2011;Li et al., 2015) or

internal solitary waves (Zhang et al., 2018;Zhao and Alford, 2006;Cai et al., 2014) generated in the LUS and propagating in the northern SCS, the SCS throughflow as a branch of the Pacific to Indian Ocean throughflow (Wei et al., 2019;Wang et al., 2011), and energetic mesoscale eddy activities (Zu et al., 2019;Xu et al., 2019;Zhang et al., 2016;Zheng et al., 2017;Hwang and Chen, 2000;Wang et al., 2020). The multi-scale dynamical mechanisms in the SCS are too complex to understand clearly as yet, it has

always been a challenge to simulate or reproduce the ocean circulations, not to mention forecast future oceanic status by Operational Oceanography Forecasting System (OOFS).

Within coordination and leadership of Global Ocean Data Assimilation Experiment (GODAE) OceanView (GOV, https://www.godae-oceanview.org; Tonani et al., 2015;Dombrowsky et al., 2009), in recent one or two decades, several regional OOFSs have been developed and operated based on the state-

of-the-art community numerical ocean models in different regions of the ocean. Tonani et al. (2015) summarized that there were 19 regional systems running operationally until 2015 in total.

For instance, Canadian Operational Network of Coupled Environmental Prediction Systems (CONCEPTS) from Canada was built based on the Nucleus for European Modelling of the Ocean (NEMO) 3.1, whose domain covered the Arctic and North Atlantic with 1/12° horizontal resolution; the

Real-Time Ocean Forecast System (RTOFS) from US National Oceanic and Atmospheric Administration (NOAA) National Centers for Environmental Prediction (NCEP) was designed based on the HYbrid Coordinate Ocean Model (HYCOM) and implemented in the North Atlantic on a curvilinear coordinate, with the resolution ranging from 4 km to 18km in horizontal; The Meteorological Research



Institute (MRI) of Japan Meteorological Agency (JMA) developed the Multivariate Ocean Variational

Estimation System/MRI Community Ocean Model (MOVE/MRI.COM) coastal monitoring and

forecasting system based on the MRI.COM (Tsujino et al., 2006). The model consists of a fine-resolution

(2km) coastal model around Japan and an eddy-resolving (10km) Western North Pacific (WNP) model

with one-way nesting; the Chinese Global operational Oceanography Forecasting System (CGOFS) was

developed and operated based on the Regional Ocean Modelling System (ROMS, Shchepetkin and

McWilliams, 2005) and NEMO by National Marine Environmental Forecasting Center, covering 6

subdomains from global to polar regions, Indian Ocean, Northwest Pacific, Yellow Sea and East China

Sea (Kourafalou et al., 2015), South China Sea (Zhu et al., 2016), with their horizontal resolutions

ranging from 1/12° to 1/30°. It is worth noting that there are considerable differences among those

systems in many aspects, such as the model codes, area coverage, horizontal/vertical resolutions, data

assimilation schemes, and so on, according to the user needs or regional ocean characteristics.

In order to better satisfy end users' needs, OOFSs need to be updated and improved continuously since

operation. In general, most improvements of OOFSs are implemented by increasing horizontal or vertical

grid resolution, or improving the data assimilation schemes from a simple one to sophisticate one in order

to assimilate more amounts and types of observation data, according to the benefits from the growth of

high-performance computing power and global or regional observation network. Initially, the

MOVE/MRI.COM was developed based on a three-dimensional variational (3DVAR) analysis scheme

and implemented in 2008 (Usui et al., 2006), then it is updated to the four-dimensional variational

(4DVAR) analysis scheme to provide better representation of mesoscale processes (Usui et al., 2017).

Mercator Ocean International (MOI) global monitoring and forecasting system had been routinely

operated in real time with an intermediate-resolution at 1/4° and 50 vertical levels since early 2001. An

upgrading of increasing horizontal resolution was implemented in December 2010, to consist a 1/12°

nested model over the Atlantic and Mediterranean. Real time daily services with a global 1/12° high-

resolution eddy-resolving analysis and forecasting were delivered by an updated system, since 19

October, 2016. Moreover, MOI also continues to implement regularly update by increasing system's

complexity, such as expanding the geographical coverage, improving models and assimilating schemes,

and have developed several versions for the various milestones of the MyOcean project and the Copernicus Marine Environment Monitoring Service (CMEMS, Lellouche et al., 2013;Lellouche et al., 2018).

As mentioned in the literature of Zhu et al. (2016), the regional SCS Operational Oceanography Forecasting System (SCSOFS, here after named it as SCSOFSv1) has been developed and routinely operated in real time since the beginning of 2013. It has continued to be upgraded by modifying model settings in many aspects, such as mesh distributions, surface atmospheric field forcing, open boundary inputs, and so on, and improving data assimilation scheme according to the results of comparing and validating from Zhu et al. (2016), in order to provide better services. The primary purpose of this paper
is to introducing updates applied to SCSOFS, but only show the highest impact on the system. The other results from routine system updates or improvements will not be illustrated or discussed in detail.

This paper is organized as follows. A detailed description of some general/basic updates applied to SCSOFS will be provided in Section 2. Some highlights and sensitive updates and their impacts to the performance of system are shown in Section 3. Results of the scientific inter-comparison and assessment
for different SCSOFS versions during the upgrading processes based on the 'Class 4 metrics' verification framework (Hernandez et al., 2009) will be shown in Section 4. Section 5 contains a summary of the scientific improvements and future plans for the next step.

## 2. Physical model description, updates and datasets

This section describes some general updates applied to the SCSOFSv1 in recent couple years. The newly
updated system is named as SCSOFSv2 here after. In order to isolate the contribution of one modification, different simulations were performed for respective updates. However,

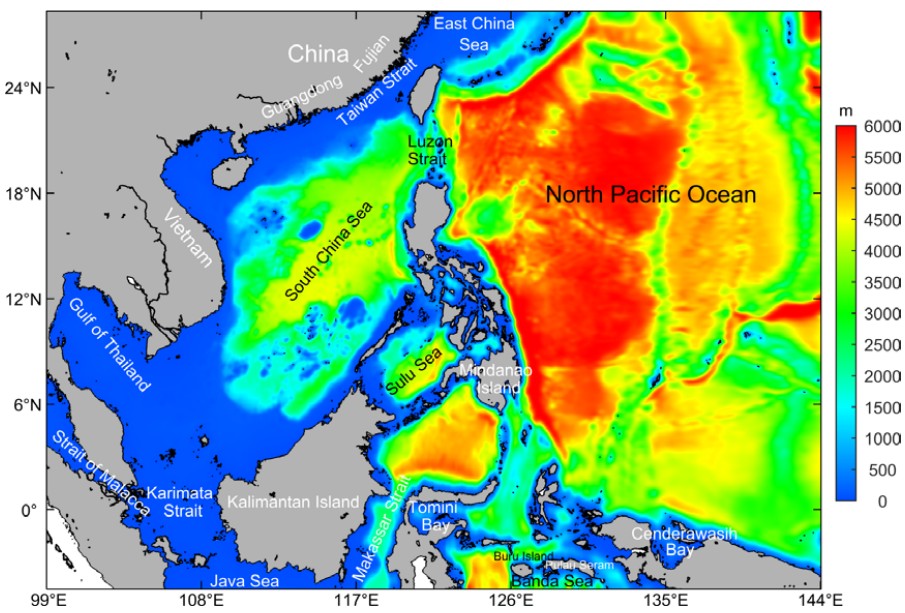

**Figure 1: The model domain and bathymetry of SCSOFSv2**

some updates have been implemented directly according to model experiences or theory knowledges,

without standalone evaluation. The performances from a few integrated updates will be shown in Section

4 for different upgrading stages.

The SCSOFSv2 is still built based on ROMS, but the version of ROMS has been updated from v3.5 (svn

trunk revision 648 in 2013) to v3.7 (svn trunk revision 874 in 2017). ROMS v3.7 incorporates some

changes for the model settings, which facilitating the operational running especially, besides of the major

overhaul of the nonlinear, tangent linear, representor, multiple-grid nesting and adjoint numerical kernels.

Firstly, we redistributed the land-sea grid mask layout to enable systems mesh land boundary fit the

actual coastline better (Fig.1). By comparing with the Fig. 1 from Zhu et al. (2016), a few sea areas had

been changed from land to sea or inverse, e.g. along the coast of China mainland, the Vietnam and the

Gulf of Thailand, around the coast of the Kalimantan Island and the Mindanao Island. In addition, the

Strait of Malacca had been opened to connect with the Karimata Strait, and the western lateral boundary

was treated as open boundary across the Strait of Malacca along 99°E, instead of a wall as in SCSOFSv1;

along the south lateral open boundary, the Java Sea was connected to the Makassar Strait in the southeast



of the Kalimantan Island, the Banda Sea was connected in the south of Buru Island and Pulau Seram; and opened the Tomini Bay and the Cenderawasih Bay. It is obvious that the land-sea masks changing

can generate important effects on the sea water volume and transport in the model domain, then would contribute to simulating more accurate ocean circulations.

The bathymetry of ETOPO1 data set, with 1 arc-minute grid resolution from U.S. National Geophysical Data Center (NGDC) is substituted by the General Bathymetric Chart of the Oceans (GEBCO_08, Grid version 20091120, http://www.gebco.net) global continuous terrain model for ocean and land with 30

arc-second spatial resolution in SCSOFSv2. It is also merged with the measured bathymetry in the area near the coast of China mainland and adjusted with the tidal range. Then it is smoothed by applying a selective filter 8 times to reduce the isolated seamounts on the deep ocean, so that the "slope parameter" $r=\mathrm{grad}(h)/h$ is lower than a maximum value $r_0=0.2$ for each grid, in order to supress the computational errors of the pressure-gradient (Shchepetkin and McWilliams, 2003). The maximum depth is set to be

6000m still, but the minimum depth to be changed from 10m in SCSOFSv1 to 5m in SCSOFSv2 (Wang, 1996). The final smoothed bathymetry is shown in Fig.1.

For the vertical terrain-following coordinate, it has been increased from 36 s-coordinate layers in SCSOFSv1 to 50 layers in SCSOFSv2. The transformation equation from the original formulation is also changed to the improved solution (Shchepetkin and McWilliams, 2005). The original vertical stretching

function (Song and Haidvogel, 1994) is replaced with an improved double stretching function (Shchepetkin and McWilliams, 2005), to make it thinner on the upper 300m in order to resolve the thermocline well.

The new initial temperature and salinity (T/S) fields in SCSOFSv2 are extracted from the Generalized Digital Environmental Model Version 3.0 (GDEMV3, Carnes, 2009) global climatology monthly mean

in January, to substitute the version 2.2.4 of Simple Ocean Data Assimilation (SODA, Carton and Giese, 2008) datasets. All four lateral boundaries are opened, whose temperature, salinity, velocity and elevation are prescribed by spatial interpolation from the new SODA 3.3.1 for the running 2005-2015 and SODA 3.3.2 for the running 2016-2018 datasets (Carton et al., 2018), instead of the original SODA 2.2.4. In this present, we use the SODA 3.3.1/2 monthly mean ocean state variables, which are mapped onto the regular



$1/2° \times 1/2°$ Mercator horizontal grid from the original approximately $1/4° \times 1/4°$ displaced pole non-Mercator horizontal grid at 50 z vertical levels.

For the surface atmospheric forcing, we changed the dataset from the NCEP Reanalysis 2 provided by the NOAA/OAR/ESRL PSD, Boulder, Colorado, USA, accessible from the website at http://www.esrl.noaa.gov/psd/ (Kanamitsu et al., 2002), to 6-hourly Climate Forecast System Reanalysis

(CFSR, http://rda.ucar.edu/datasets/ds093.0, Saha et al., 2010) for 2005-2011 and Climate Forecast System Version 2 (CFSv2, http://rda.ucar.edu/datasets/ds094.0, Saha et al., 2014) for 2011-2018. Both of them are archived at the National Center for Atmospheric Research (NCAR), Computational and Information Systems Laboratory, Boulder, Colorado, with a 0.2°-0.3° significantly higher horizontal grid than the $2.5° \times 2.5°$ resolution for NCEP Reanalysis 2.

The net surface heat flux correction is still following Barnier et al. (1995) in SCSOFSv2, but the parameter (dQ/dSST) of kinematic surface net heat flux sensitivity to sea surface temperature (SST) is calculated using SST, sea surface atmospheric temperature, atmospheric density, wind speed and sea level specific humidity, instead of setting a constant number -30 W $m^2$ $K^{-1}$ for the whole domain as in SCSOFSv1. So the parameter dQ/dSST varies temporally and spatially. Meanwhile, we also replace the

merged satellite's infrared sensors and microwave sensor, and *in-situ* (buoy and ship) data global daily SST (MGDSST) obtained from the Office of Marine Prediction of the Japan Meteorological Agency (JMA), with the infrared Advanced Very High Resolution Radiometer (AVHRR) satellite data, which is an analysis constructed by combining observations from different platforms on a regular grid via optimum interpolation and provided by National Centers for Environmental Information (NCEI).

The North Equatorial Current (NEC) is an interior Sverdrup steady current in the subtropical NP and located at about 10°N-20°N, and usually bifurcates into two branches after encountering the western boundary along the Philippine coast in the west of 130°E (Qiu and Chen, 2010). However, the NEC is separated into two branches since the model's eastern lateral boundary in SCSOFSv1, its main branch located at about 9.5°N-13°N, the other branch located at 14.5°N-17°N (Fig. 2a), which is clearly not in

line with the fact. This is because the Guam Island (Fig. 2, red circle) which located about (13°26′N, 144°43′E) is included in SCSOFSv1 and whose location is too close to the eastern lateral boundary. To

resolve this problem, the eastern lateral boundary has been moved westward from 145°E to 144°E to

narrow the model domain and exclude the Guam Island in SCSOFSv2. It is found that the simulated NEC

keeps the form of one main current until 130°E,

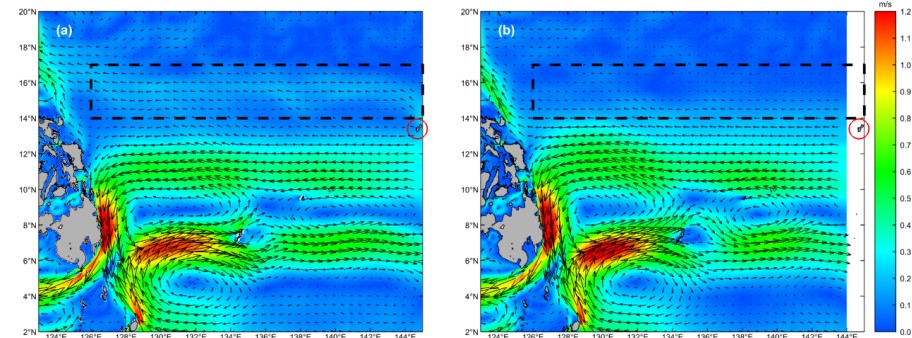

**Figure 2: The multi-year monthly mean sea surface currents (color shaded for current speed (m s-1), arrows for current direction) with vertical averaged above 100m in May. The left panel is from SCSOFSv1 with the model domain including the Guam Island, the right panel is from SCSOFSv1.2 with the eastern lateral boundary moving westward.**

then bifurcates into the southward-flowing Mindanao Current and the northward-flowing Kuroshio in

SCSOFSv2 (Fig. 2b).

The SCSOFSv2 is run with 5s time step for the external mode, and 150s for the internal mode under all

new configurations mentioned above and in Section 3. The modification for the time step is due to the

change of the discrete schemes, which would be illustrated in Section 3. Before the operational run, a 26

years climatology run is conducted for spinning-up, and is followed with a hindcast run from 2005 to

2018 (Wang et al., 2012). The daily mean of model results is archived and used to validate in the

following parts of this paper.

## 3. Highlights and sensitive updates and their impacts

Most of bias or errors in the operational systems mainly induced by initial errors and model deficiencies,

which can be attributed to some major recurring problems like sea surface atmospheric forcing, intrinsic

deficiencies of numerical model (e.g. discrete schemes, parameterization schemes for sub-grid scale),





and the assimilation schemes. In this section, we elaborate solutions, that are not mentioned in Sect.2, to
such problems applying in SCSOFSv2.

### 3.1 Sea surface atmospheric forcing

The air-sea interaction is one of essential physical processes that affect vertical mixing and thermal
structure of the upper-ocean. The air-sea fluxes mainly include momentum flux, fresh water flux and
heat flux. SST is an important indicator of ocean circulation, ocean front, upwelling and sea water mixing,
whose variation mainly depending on the air-sea interaction and the ocean thermal and dynamical factors
(Bao et al., 2002). Thus, for OOFS and ocean numerical modelling, accurate simulation and forecasting

of SST is one of the most important metric to evaluate the modelling and forecasting skill.

The accurate input of sea surface atmospheric forcing plays a key role to excel in model simulation of
SST. ROMS provides two methods to introduce sea surface atmospheric forcing: the first option is
directly forcing ocean model by providing momentum fluxes (wind stress), net fresh water fluxes, net
heat fluxes and shortwave radiation fluxes from atmospheric datasets; the second is employing the Fairall

et al. (2003) COARE3.0 bulk algorithm to calculate air-sea momentum, fresh water and heat turbulent
fluxes using the set of atmospheric variables from atmospheric datasets including wind speed at 10m
above sea surface, mean sea level air pressure, air temperature at 2m above sea surface, air relative
humidity at 2m above sea surface, downward longwave radiation flux, precipitation rate and shortwave
radiation fluxes (Large and Yeager, 2009). Since the SST using in the calculation of those three air-sea

fluxes is extracted from ocean model, the increasing of SST thus induces the variations of sensible heat
flux, latent heat flux, and longwave radiation then increasing loss of ocean heat, and inhibiting the further
increasing of SST, and vice versa. It means that an effective negative feedback mechanism could form
between SST and SST-related heat fluxes. In this case, the simulated SST would maintain at a reasonable
level. The first one is employed in SCSOFSv1, the second one, bulk algorithm, is employed in

SCSOFSv2.

In order to evaluate the performances of different sea surface atmospheric forcing methods, we conduct
a special experiment by changing the method based on SCSOFSv1, here named the experiment as
BulkFormula. In this experiment, we use the merged satellite SST analysis with a multi-scale optimal




interpolation called the Operational SST and Sea Ice Analysis (OSTIA) system, which globally coverage

on a daily basis at a horizontal grid resolution of 1/20° (~6 km) and provided by the Met Office (Donlon

et al., 2012), to verify the results of SCSOFS.

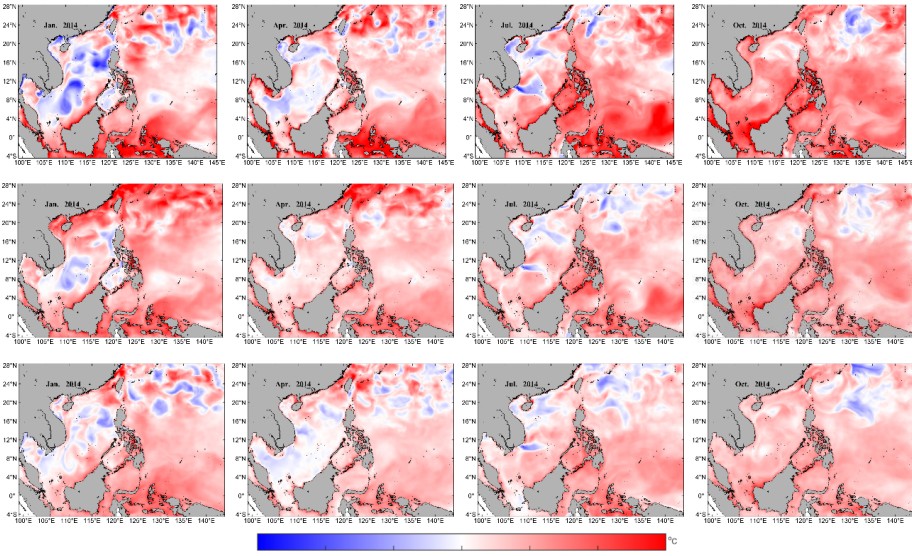

**Figure 3: The monthly mean SST differences in January, April, July, and October, 2014: SCSOFSv1 minus OSTIA SST (upper panels), BulkFormula minus OSTIA SST (middle panels), SCSOFSv2 minus OSTIA SST**
**(lower panels)**

Figure3 shows the distributions of monthly mean SST differences of SCSOFSv1, BulkFormula,

SCSOFSv2 minus OSTIA SST in January, April, July and October, 2014 to stand for Winter, Spring,

Summer and Autumn, respectively. It is found that the simulated SST are higher than OSTIA SST for all

three results in general. The differences are pronouncedly higher for the results from SCSOFSv1 than

the results from BulkFormula and SCSOFSv2. The maximum differences mainly occur near coast (Fig.3

upper panels), especially for a few bays embedded into the mainland which is hard to be resolved well

with 2-3 horizontal grids at 1/30° resolution and in very shallow water depth in SCSOFSv1. This is

because sea surface atmospheric forcing data is not accurate enough near the coast, and provide

abnormally more heat to ocean causing the continuously heating up of ocean. Thus, simulated SST is

beyond normal level in SCSOFSv1. This phenomenon can be alleviated significantly by introducing the



effective negative feedback mechanism between model's SST and air-sea heat flux by employing the

COARE 3.0 bulk algorithm in both BulkFormula and SCSOFSv2 (Fig.3 middle and lower panels).

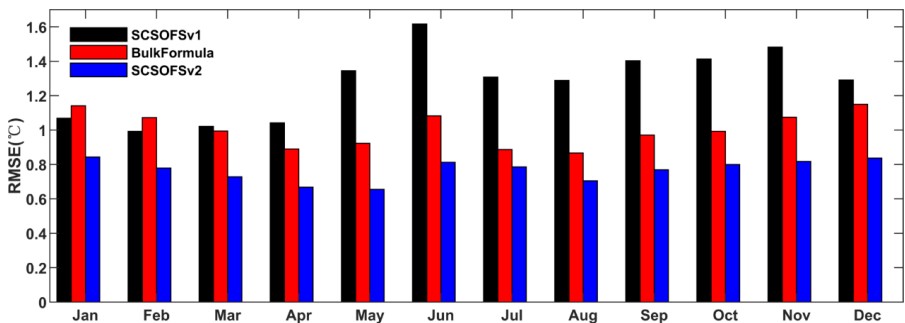

**Figure 4: Domain averaged monthly mean SST RMSE comparison among SCSOFSv1(black), BulkFormula (red), SCSOFSv2 (blue) and OSTIA SST in January, April, July, and October, 2014**

Figure 4 shows histograms of domain averaged Root-Mean-Square Error (RMSE) of monthly mean SST differences of SCSOFSv1, BulkFormula, SCSOFSv2 with respect to OSTIA datasets in each month of 2014. It is found that the domain averaged RMSE of monthly mean SST differences from SCSOFSv1 is about 0.99-1.62°C, the annual mean value is about 1.27°C. The highest (1.62°C) is in June, the lowest (0.99°C) is in February. Monthly mean RMSE for BulkFormula run is about 0.87-1.15°C, the annual mean value is about 1.00°C, the maximum value (1.15°C) is in January and December, the minimum value (0.87°C) is in August. The performance of model skill for the annual mean SST RMSE can be improved by about 21% only by changing the method of sea surface atmospheric forcing from directly forcing to COARE 3.0 bulk algorithm, due to effective negative feedback mechanism.

**3.2 Tracers advection term discrete schemes**

Spurious diapycnal mixing is one of traditional errors in state-of-the-art atmospheric and oceanic model, especially for the terrain-following coordinate regional models including both the continental slope and deep ocean (Marchesiello et al., 2009; Naughten et al., 2017; Barnier et al., 1998). Marchesiello et al. (2009) identified the problem of the erosion of salinity from the southwest Pacific model with steep reef slopes and distinct intermediate water masses based on ROMS. They found that ROMS cannot preserve the large-scale water masses while using the third-order upstream advection scheme during the spin-up





phase of the model, and proposed a rotated split upstream third-order scheme (RSUP3) to decrease

dispersion and diffusion by splitting diffusion from advection. They implemented RSUP3 by employing

a rotated biharmonic diffusion scheme with flow-dependent hyper diffusivity satisfying the Peclet

constraint.

For SCSOFSv1, a third-order upstream horizontal advection scheme (hereafter referred to as U3H), a

fourth-order centered vertical advection scheme (hereafter referred to as C4V), and the scheme of

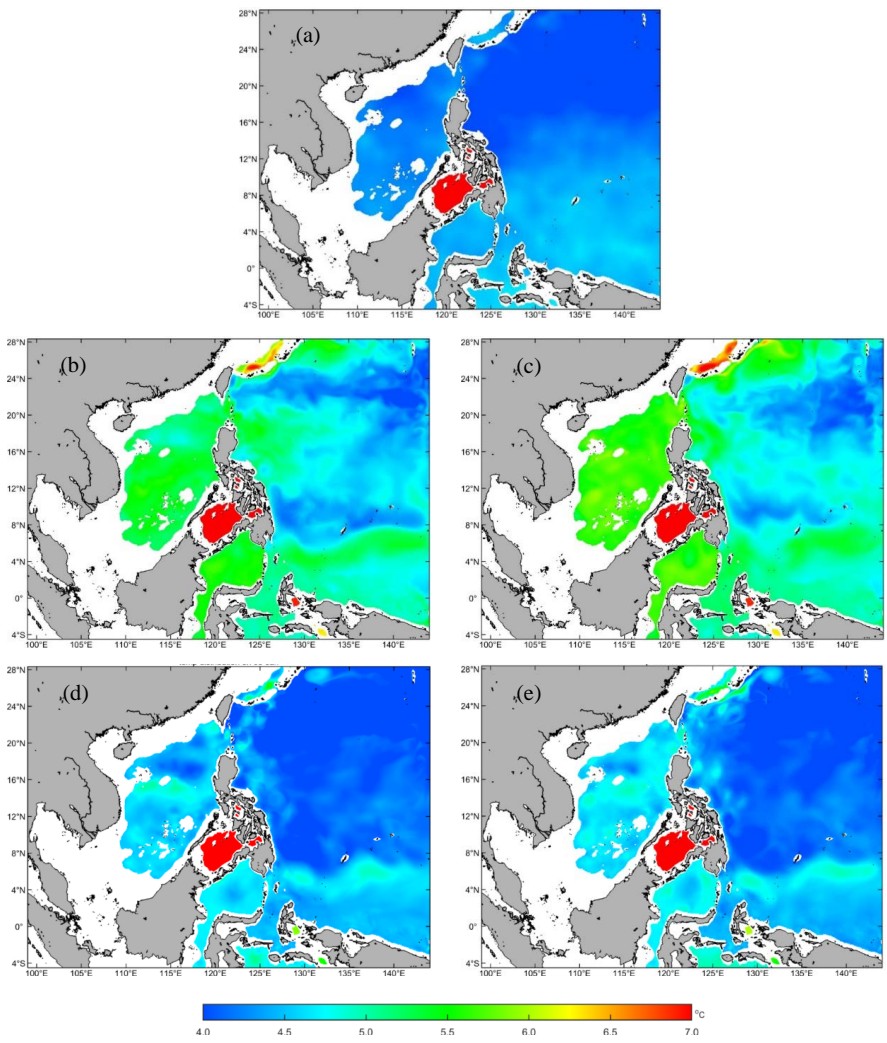





**Figure 5: The distributions of monthly mean temperature at 1000m layer in January from GDEMv3**
        **climatology (a), the fifth (b) and the eleventh (c) model year by using the scheme combination of UCI based**
        **on SCSOFSv2 for other model settings, the fifth (d) and the eleventh (e) model year by using the scheme**
        **combination of AAG based on SCSOFSv2 for other model settings.**

        horizontal mixing on epi-neutral (constant density, hereafter referred to as ISO) surfaces for tracers are

        selected. We have encountered same problem with Marchesiello et al. (2009) for temperature (Fig.5b

and 5c) and salinity (Fig.6b and 6c). Figure 5 and 6 show the distributions of monthly mean temperature

        and salinity at 1000m layer in January from GDEMv3 climatological initial fields, and the simulated

        results from the fifth and the eleventh model years by using the scheme combination of U3H, C4V and

        ISO (hereafter referred to as UCI) and the combination of the fourth-order Akima scheme (Shchepetkin

        and McWilliams, 2005) for both horizontal and vertical advection terms and the scheme of horizontal

mixing on geopotential surfaces (constant Z) for tracers (hereafter referred to as AAG), respectively, and

        other settings are identical with SCSOFSv2. Figure 7 shows the comparisons of time series of domain

        averaged monthly mean temperature and salinity at 1000 m layer simulated using the scheme

        combinations of UCI and AAG based on SCSOFSv2, respectively. In order to save computation costs,

        we only run the model with scheme combination of UCI for over 16 years till it is in stable status.

During the spin-up phase of the model from the initial conditions derived from GDEMV3, the

        temperature at 1000 m increases from 3.0-12.0°C in initial (Fig.5a) to 3.0-17.2°C (Fig.5b), and the

        domain averaged monthly mean value quickly increases from 4.4 °C in initial to 5.1 °C (Fig.7a) in

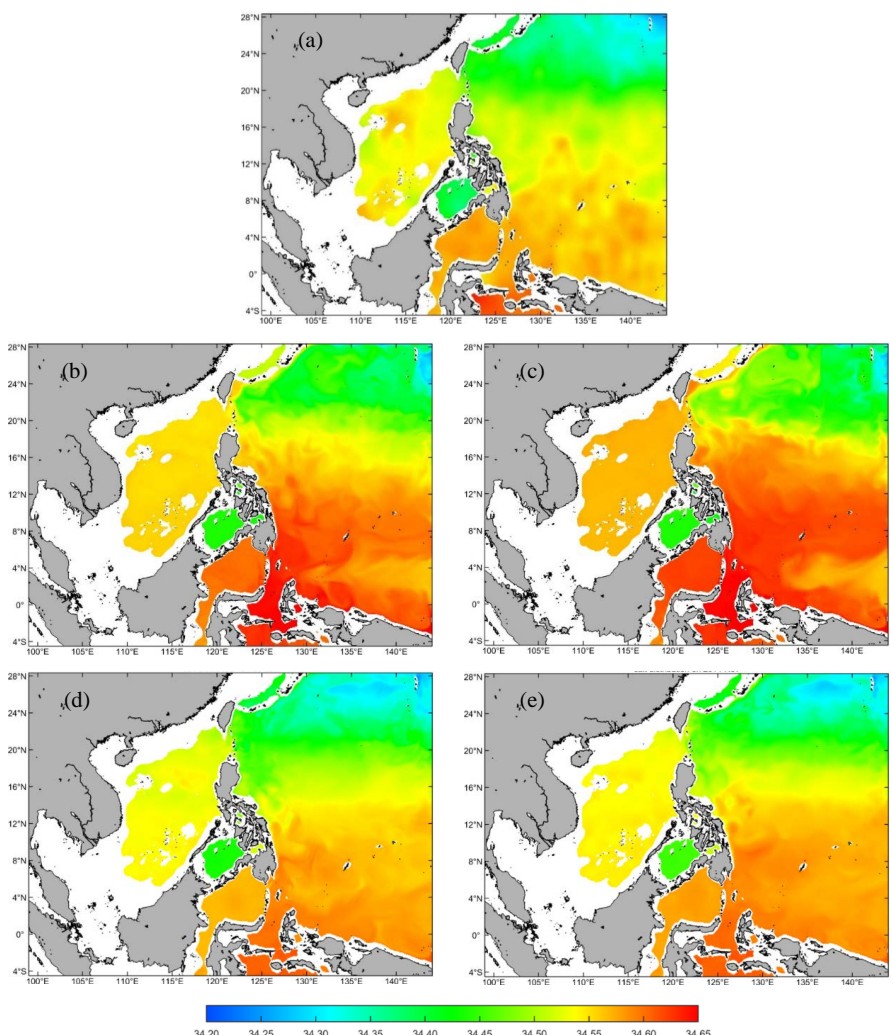

**Figure 6: The same as Fig.5, but for salinity.**

January of the fifth model year; the salinity at 1000 m increases from 34.26-34.62 in initial (Fig.6a) to

34.27-34.68 (Fig.6b), and the domain averaged monthly mean value increases rapidly from 34.50 in

initial to 34.54 (Fig.7b) in January of the fifth model year too. Especially, the increasing of domain

averaged monthly mean value is almost linearly for both temperature and salinity in the first 50 months,

indicating a fast increasing speed and strong spurious diapycnal mixing (Fig.7). Those values are even

higher in January of the eleventh model year, the ranges (minimum and maximum value) reach to 3.0-

17.3°C for temperature (Fig.5c) and 34.26-34.73 for salinity (Fig.6c). The domain averaged values are

5.3 °C for temperature and 34.56 for salinity (Fig.7), respectively. The areas with increasing temperature

and salinity are mainly located at steep slopes and nearby regions, e.g. the central basin of SCS, the

Sulawesi Sea and the equatorial Pacific Ocean.

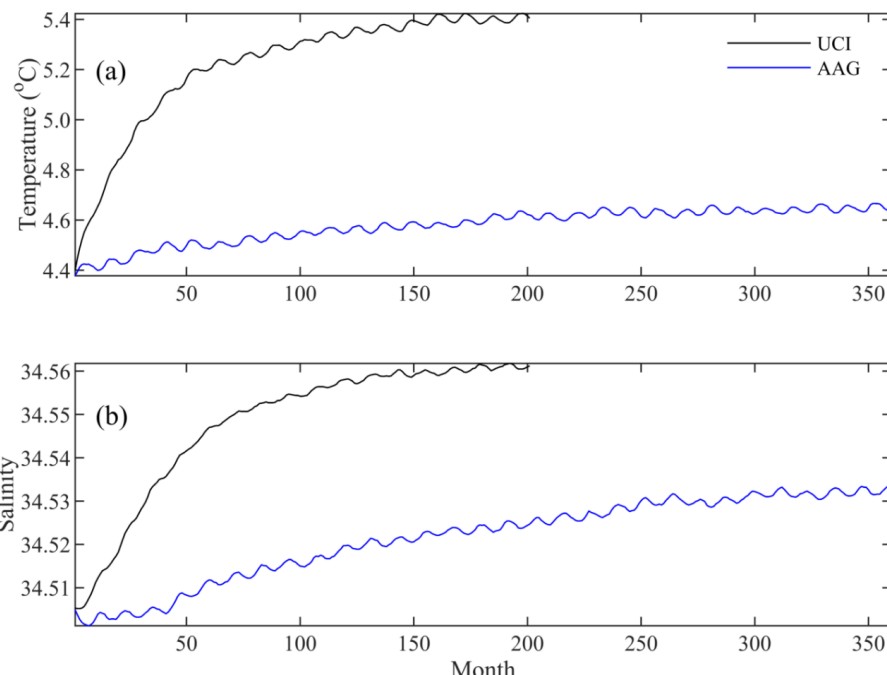


**Figure 7: The timeseries of domain averaged monthly mean temperature (a) and salinity (b) at 1000 m layer
simulated by using the scheme combinations of UCI (black line) and AAG (blue line), respectively**

To fix this problem, we tested various model settings and compiling options available in ROMS, such as

increasing the number of vertical levels, changing the advection and diffusion schemes, horizontal

mixing surfaces for tracers, horizontal mixing schemes. Details of how tested model settings effect on

the spurious diapycnal mixing are beyond the scope of this paper, which will be discussed in a separate

paper. Based on test results, we conclude that the spurious diapycnal mixing problem can be suppressed

significantly by employing AAG scheme combination (Fig.5d, e and Fig.6d, e) in SCSOFSv2.

The monthly mean temperature at 1000 m layer from SCSOFSv2 varies from 3.0-12.0°C in initial

condition to 3.0-11.5°C (Fig.5d), and the domain averaged monthly mean value increases slightly from

4.4 ℃ in initial to 4.5 ℃ (Fig.7a) in January of the fifth model year. The salinity at 1000 m varies from 34.26-34.62 in initial condition to 34.24-34.63 (Fig.6d), and the domain averaged monthly mean value only slightly varies from 34.505 in initial to 34.509 (Fig.7b) in January of the fifth model year. Those values show little variation till January of the eleventh model year, the ranges are 3.0-11.3℃ for

temperature (Fig.5e) and 34.25-34.63 for salinity (Fig.6e), and the domain averaged values are 4.6 ℃ for temperature and 34.52 for salinity (Fig.7), respectively. For the increment of domain averaged values, temperature is about 0.2℃ and salinity is about 0.03, yet remaining stable after 20 model years (Fig.7). It is suggested that spurious diapycnal mixing has been suppressed significantly by AAG scheme combination, which can preserve the characteristics of water masses in deep ocean well.

In addition, it is found that the model skill for SST has also been improved significantly while the new AAG scheme employed in SCSOFSv2 (Fig.3 and Fig.4). The maximum of monthly mean differences between simulated SST by SCSOFSv2 and OSTIA is about 3-4℃, which is smaller than the results from BulkFormula obviously. The results of SCSOFSv2 show much more areas with lower SST than OSTIA in the middle of the Pacific Ocean, comparing to the results of SCSOFSv1 and BulkFormula, which can

be attributed to the new scheme combination. For the domain averaged RMSE of monthly mean SST of SCSOFSv2 is about 0.65-0.84℃, with an annual mean value of 0.77℃, the maximum value (0.84℃) is in January and December, the minimum value (0.65℃) is in May. Comparing with the results of BulkFormula, the performance of model skill based on the annual mean SST RMSE is improved by about 23% due to employing new combination scheme in SCSOFSv2.

**3.3 Data assimilation scheme**

As mentioned as Zhu et al. (2016), the original SCSOFSv1 had employed the multivariate Ensemble Optimal Interpolation (EnOI, Evensen, 2003;Oke et al., 2008) method to assimilate the along track altimeter Sea Level Anomaly (SLA) data produced by SSALTO/DUACS and distributed by AVISO with support from Center National D'études Spatiales. During this upgrading process, we also improved some

functions of EnOI scheme, and developed a new "Multi-source Ocean data Online Assimilation System" (MOOAS).

Ocean Science Discussions Open Access EGU

Firstly, SCSOFSv1 assimilated the along track SLA data only, while SCSOFSv2 is able to assimilate, in addition, satellite AVHRR SST and in-situ temperature and salinity (T/S) vertical profiles data from the Argo arrays, simultaneously. It is conducted by constructing all innovations (difference between the assimilated observation and the model forecast), background error covariance, and observation errors for four different variables to one array, respectively.

Secondly, we have introduced the method of computing the anomalies used to construct the background error covariance following Lellouche et al. (2013). In SCSOFSv1, the anomalies are computed from a long-term (typically 10 years) numerical simulation without assimilation with respect to the temporal mean of the whole time series for the ocean state, i.e. sea surface height and three-dimensional temperature, salinity, zonal velocity, and meridional velocity. The snapshots with 5-day interval are archived and used as the ensemble members in SCSOFSv1. However, in SCSOFSv2, a Hanning low-pass filter is employed to create running mean according to Lellouche et al. (2013). Thus the anomalies are computed from a 10-year (2008-2017) free run with respect to the running mean in order to estimate the 20-day (temporal window for running mean) scale error on the ocean state at 60-day period of each year in our case. Especially, it is pointed out that the daily averaged free run results are archived and used to be ensemble members in SCSOFSv2. It means that the background error covariances rely on a fixed basis and sub-seasonally variable ensemble of anomalies, which improves the dynamic dependency.





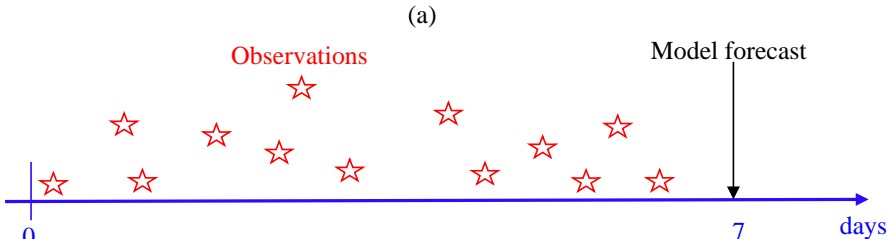

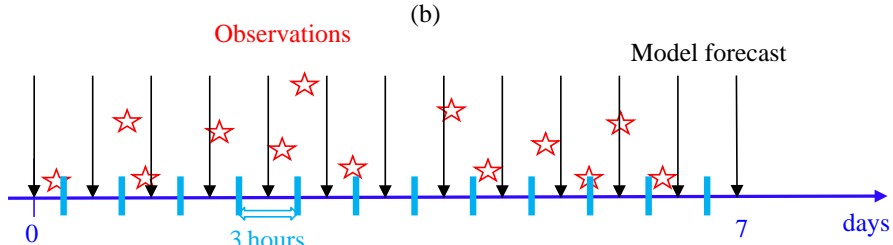

**Figure 8: Schematic representation of the FGAT method not used in SCSOFSv1 (a) and used in SCSOFSv2 (b). Red stars stand for observations, black arrows stand for archived snapshots of model forecast**

Thirdly, for each analysis step with a 7-day assimilation cycle, all observations of SLA within the 7-day time window before the analysis time are treated as observed at the analysis time in SCSOFSv1, with assuming that all observations are still valid at the analysis time. The time misfit between the observation and model forecast would cause non-negligible biases for the calculation of innovations. Actually, it is close to impossible to calculate the synchronous innovations between the observation and model forecast entirely, since the temporal distributions of SLA and Argo data are irregular and variable at each analysis step. In order to alleviate this deficiency, the First Guess at Appropriate Time (FGAT) method (Lee and Barker, 2005;Cummings, 2005;Lee et al., 22-25 June 2004;Sandery, 2018) is used in SCSOFSv2. Considering the intense computing and storage cost, we have divided the 7-day time window into 56 3-hour time slots (Fig.8), and archived 57 snapshots with a 3-hour interval while model forecast running following the previous analysis. Then the innovations can be calculated within each 3-hour time slot by using the observations minus the nearest model forecast. It means that the maximum time misfit of the

innovations between the observation and model forecast would be decreased from 7 days to 1.5 hours by

using FGAT.

In SCSOFSv1, the analysis increments of sea surface height and three-dimensional temperature, salinity,

zonal and meridional velocities produced by each analysis of data assimilation are applied to the model

initial fields at one time step. It would induce model significant initial shock and spurious high-frequency

oscillation due to the imbalance between the increments and the model physics inevitably (Lellouche et

al., 2013; Ourmières et al., 2006), and usually causes a rapid growth of forecast error and even model

blow up after a few assimilation cycles or one or two-year period after the intermittent assimilation run.

It is a threat to the stability and robustness of OOFS. Therefore, we introduced the incremental analysis

update (IAU) method (Bloom et al., 1996; Ourmières et al., 2006) to apply each analysis increment to

the model integration as a forcing term in a gradual manner in SCSOFSv2 to diminish the negative impact.

In our case, we get the tendency term by dividing the increments with the total number of time steps

within an assimilation cycle as in most IAU methodologies, in order to make sure the time integral of

tendency term equals the analysis increment calculated by EnOI.

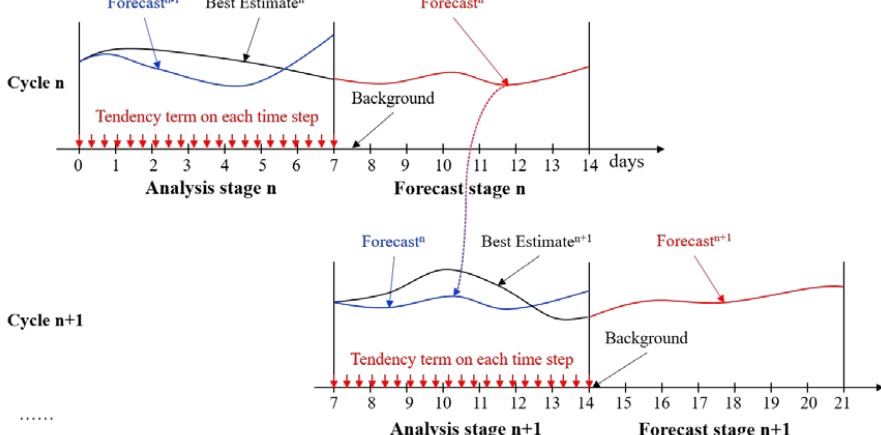

**Figure 9: Schematic representation of the data assimilation procedure for two consecutive cycles, n and n+1**
**in SCSOFSv2, while considering the FGAT and IAU methods.**


Once including the FGAT and IAU methods in EnOI scheme, the whole system integral strategy has to be adjusted by adding one more model integration over the assimilation time window (Lellouche et al., 2013). In SCSOFSv1, only one time model integration is needed. It means that once physical ocean model finish 7 days run (does not need to archive snapshot fields) and output a restart field, the EnOI

data assimilation module is started to calculate the analysis increments at the restart field time and add it to the restart field, then the physical ocean model makes a hot-start from the updated restart field to run 7 days for next cycle.

However, in SCSOFSv2, two times model integration is needed due to considering the FGAT and IAU methods (Fig.8). It means that physical ocean model needs to be integrated 14 days in each assimilation

cycle, to add the tendency term to the model prognostic equations due to the IAU method used during the first 7 days run (referred to as "Analysis Stage"), to output restart field at the end of 7th day for hot starting ocean model in next cycle, and to output 3-hourly snapshots forecast fields during the second 7 days run (referred to as "Forecast Stage") to be used in next cycle by FGAT method. The model outputs from the Analysis Stage are referred to as "Best Estimate", and from the Forecast Stage are referred to

as "Forecast". The analysis increments are defined at the 3.5th day, but not at the end of 7th day as in SCSOFSv1, with the observed SLA and Argo T/S vertical profiles data within the 7-day time window and AVHRR SST data on the 4th day used by FGAT method.

### 4.    Scientific inter-comparison and accuracy assessment

In order to show the improvements of different SCSOFS sub-versions during the upgrading process, the

results of scientific inter-comparison and assessment are shown in this section, by using the GOV Inter-comparison and Validation Task Team (IV-TT) Class 4 verification framework (Hernandez et al., 2009). Class 4 metrics are used for inter-comparison and validation among different global or regional OOFSs or assimilation systems originally (Ryan et al., 2015; Hernandez et al., 2015; Divakaran et al., 2015). It includes four metrics, namely, bias for consistency, RMSE for quality or accuracy, Anomaly Correlation

(AC) for pattern of the variability and skill scores for the utility of a forecast. They are calculated according to differences between model values and reference measurements in observations space for





each variable over a given period and spatial domain. The physical variables used in Class 4 metrics are SST, SLA, T/S profiles, surface currents and sea ice. Reference measurements, providing ocean "truth", are selected as follow, SST from *in-situ* drifting BUOY, SLA from AVISO along-track data, temperature

and salinity from Argo profiles, respectively. They are assembled by GOV IV-TT participating partners on a daily basis (Ryan et al., 2015).

It is virtually impossible to test and validate exhaustively performances of all upgrades mentioned in Section 2 and 3. Here, we separate the whole procedure from SCSOFSv1 to SCSOFSv2 into four stages with three more sub-versions (v1.1, v1.2, v1.3) according to the reality. By respecting to the previous

version, the major upgrades in each new version are listed in Table1.

Table 1 The major upgrades with respect to the previous version

| SCSOFS versions | Settings updates |
|---|---|
| v1→v1.1 | **ROMS version** shifting from v3.5 to v3.7; **land-sea mask** redistribution; **bathymetry** substitution ETOPO1 with GEBCO_08; **initial T/S conditions** changing from SODA2.2.4 to GDEMV3; **open boundary data** changing from climatological monthly mean to monthly mean from 1990 to 2008 with SODA 2.2.4; **sea surface atmospheric forcing data** changing from NCEP Reanalysis 2 to CFSR; **the parameter dQ/dSST** changing from constant to temporal and spatial varying values; **sea surface atmospheric forcing method** changing from directly fluxes forcing to BulkFormula |
| v1.1→v1.2 | **Open boundary data** of SODA 2.2.4 monthly mean extending from 2008 to 2010; **the eastern lateral boundary** moving westward; **the observed SST data** for net surface heat flux correction changing from MGDSST to AVHRR |
| v1.2→v1.3 | Considering **mean seal level atmospheric pressure** effect, increasing **vertical layers** from 36 to 50; changing **the transform and stretching function**; **tracers advection discrete schemes** changing from UCI to AAG |
| v1.3→v2 | Including the MOOAS |





In this paper, we use Class 4 metrices and select the first four physical variables, SST, SLA and T/S, to inter-compare and assess the accuracy among different sub-versions of SCSOFS (Table 2). Since all the reference measurements data mentioned above have not been used in SCSOFS for those sub-versions

without data assimilation, they are independent reference observation from SCSOFS except for SCSOFSv2. The inter-comparison and validation among those sub-versions without data assimilation are conducted for the model free-run results in 2013, and between v1.3 and v2 are conducted in 2018 to validate the performance of MOOAS.

**Table 2 Mean values of each metric of the four physical variables for the best estimates of each sub-version**

| Variables | Metrics | v1 | v1.1 | v1.2 | v1.3 | | v2 |
|---|---|---|---|---|---|---|---|
| **SST** | AC | 0.52 | 0.56 | 0.58 | 0.62 | 0.64 | 0.74 |
| | Bias(°C) | 0.77 | 0.88 | 0.70 | 0.40 | 0.34 | 0.24 |
| | RMSE(°C) | 1.21 | 1.12 | 0.98 | 0.76 | 0.66 | 0.52 |
| **SLA** | AC | — | — | — | — | 0.67 | 0.85 |
| | Bias (cm) | -7.0 | -5.5 | -7.0 | -7.4 | -5.2 | -3.1 |
| | RMSE (cm) | 21.6 | 20.8 | 16.7 | 14.8 | 12.9 | 8.5 |
| **T Profile** | AC | 0.01 | 0.04 | -0.12 | 0.48 | 0.38 | 0.57 |
| | Bias (°C) | 0.98 | 0.75 | 0.30 | -0.15 | -0.08 | 0.15 |
| | RMSE (°C) | 1.75 | 1.60 | 1.44 | 1.03 | 0.96 | 0.67 |
| **S Profile** | AC | -0.01 | -0.02 | 0.02 | 0.44 | 0.30 | 0.51 |
| | Bias | 0.06 | 0.05 | 0.06 | 0.02 | 0.013 | 0.009 |
| | RMSE | 0.14 | 0.14 | 0.13 | 0.10 | 0.11 | 0.08 |
| **Year** | | | 2013 | | | 2018 | |

**4.1  SST**

For the whole improving process, the accuracy of SST is continuously increasing from version v1 to v2, with AC increasing from 0.52 in v1 to 0.74 in v2 with percentage increase (PI) being 29.7%, RMSE decreasing from 1.21°C in v1 to 0.52°C in v2 with PI being 57.0%, for the annual mean of whole model domain averaged in 2013 (or v1.3 and v2 in 2018) (Table 2). For the versions v1, v1.1, v1.2, v1.3, their

AC show significant seasonal variations, with high AC in summer and low AC in winter. It is also indicated that the accuracy of SST can be benefited from the sea surface atmospheric forcing method, more accurate observed SST data used for sea surface heat flux correction, temperature advection discrete scheme, and SST data assimilation.



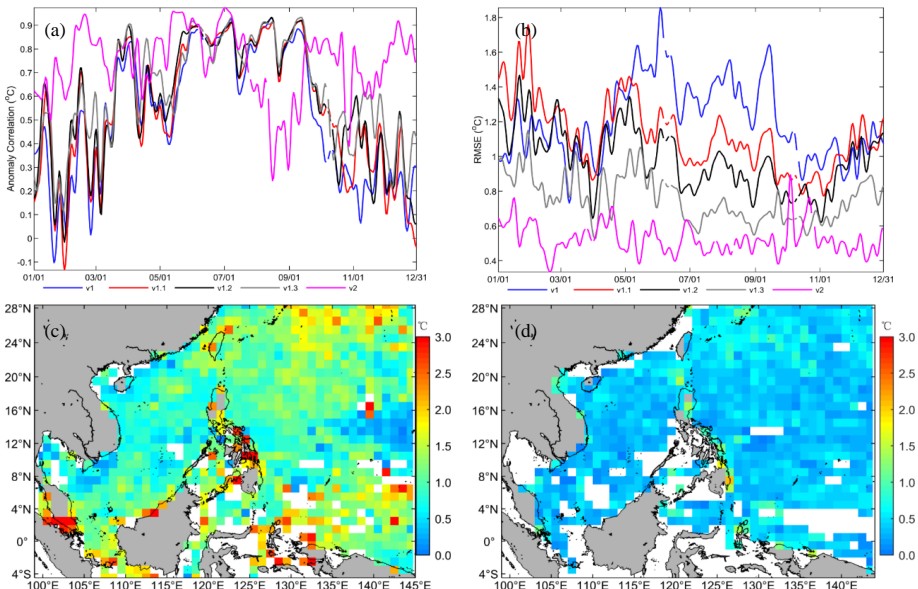

**Figure 10: Anomaly correlation (a) and RMSE (b) timeseries of SST best estimates for each version against observations as a function of time (7-day low pass filter applied), v1, v1.1, v1.2, v1.3 without data assimilation in 2013, and v2 with data assimilation in 2018. Horizontal distribution of SST RMSE in a 1°×1°bin for the version v1 (c) and v2(d), the calculation was performed for year round in 2013 and 2018, respectively**

The improvement of SST due to sea surface atmospheric forcing method changing mainly occurred in summer time, showing the same pattern as the result of the year 2014 in Fig. 3 and 4. But sea surface heat flux correction with more accurate observed SST data can improve accuracy of SST for the whole year (v1.2 in Fig.10b). We also found that OISST data is closer to OSTIA than MGDSST (figure not shown). Due to benefit from those changes, the maximum and minimum value of SST RMSE have decreased from 1.92°C and 0.71°C in v1 to 1.52°C and 0.60°C in v1.2 for the whole year 2013, respectively. It is noteworthy that AAG schemes combination not only improves the deep layer temperature, but also contributes to the improvement of SST due to internal baroclinic vertical heat transport. The maximum and minimum value of SST RMSE is 1.21°C and 0.52°C in v1.3. For the results with data assimilation in v2, the maximum and minimum value of SST RMSE is only 1.13°C and 0.32°C, respectively. It is better than the result in v1.3 year-round.



For the horizontal distribution of SST RMSE, large values are mainly located at the areas near equator,

coast areas and northern lateral boundary, with most of values larger than 1.5℃ and maximum value

about 6.67℃ in v1 (Fig.10c). In v1.3, due to the contributions of all the above model updates, the pattern

of RMSE is similar with v1 basically without significant variations, but the maximum value decreases to

3.91℃ and most of values are smaller than 1.2℃. After applying MOOAS in v2 (Fig.10d), only a few

large RMSE values are located at the eastern coast of Philippine island with the maximum value of 2.09℃

and most of values lower than 0.8℃. It indicates that the performance of SST in SCSOFSv2 has been

improved significantly due to all the updates mentioned above.

## 4.2 SLA

For the whole improving process, the accuracy of SLA is also continuously increasing from version v1

to v2, with RMSE decreasing from 21.6cm in v1 to 8.5cm in v2 with PI being 60.6%, for the annual

mean of whole model domain averaged in 2013 (or in 2018 for v1.3 and v2) (Table 2). Since there was

an ongoing problem with the SLA climatology variable provided by GODAE IV-TT during 2013-2015,

we could not calculate AC for SLA in 2013 and had feedbacked this issue to GODAE IV-TT. But from

the result of SLA AC in 2018, we can find that it increases from 0.67 in v1.3 to 0.85 in v2, showing

significant improvement for the correlation of pattern of the variability between the model results and

climatology.

From Fig.11(a), there is a slightly decrease of RMSE in v1.1 with respect to v1, which mainly occurs in

winter time, and rarely in summer time. This may because there is no direct or intrinsic relationship

between those model updates from v1 to v1.1 and SLA in physics, and those updates mainly focus on

the data sets' horizontal and temporal resolution. However, the improvement of SLA accuracy is obvious

in v1.2 with respect to v1.1, with the minimum and maximum of daily-mean RMSE values change from

0.12cm and 0.31cm in v1.1 to 0.11cm and 0.23cm in v1.2, respectively. Their annual mean value

decreases from 20.8cm in v1.1 to 16.7cm in v1.2, with PI of 19.7%. This may be resulted from good

represents of NEC pattern due to change of model eastern lateral boundary. With respect to v1.2,

accuracy of SLA in v1.3 slightly increases with annual mean value 14.8cm and PI 11.4%. It may be



resulted from the mean sea level air pressure correction and modification of T/S baroclinic structures due

to AAG employed. In addition, the most significant improvement for SLA is introduced by MOOAS,

with minimum and maximum of daily-mean RMSE value are 6.1cm and 12.1cm in v2, respectively. The

annual mean RMSE decreases to 8.5cm and PI reaches to 34.1% with respect to v1.3 and to 60.6% with

respect to v1. It is undoubtedly that this significant improvement is introduced by along-track SLA being

assimilated into the system by MOOAS.

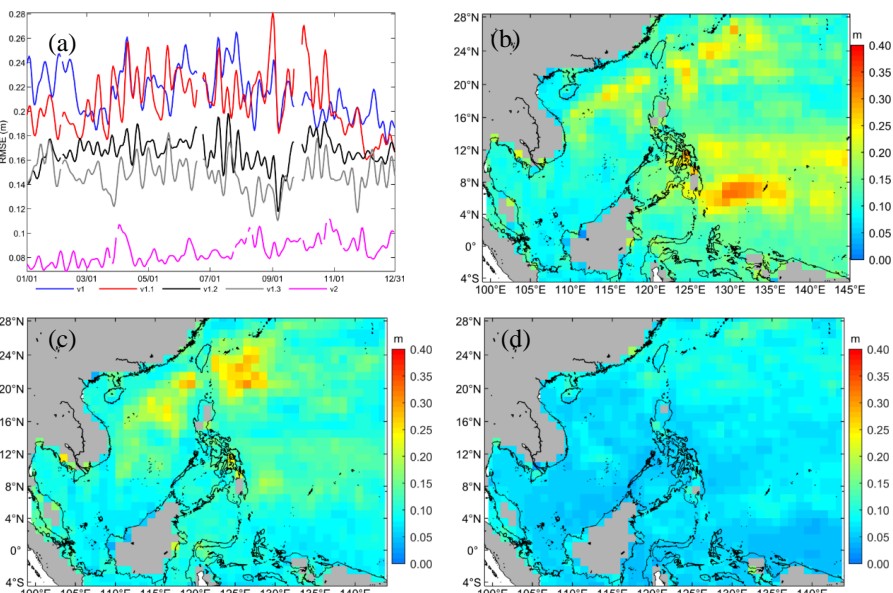

**Figure 11: (a) similar to Fig.10(b) but for SLA. (b), (c), (d) similar to Fig.10(c) or (d), but for SLA in v1, v1.3(in 2013) and v2, respectively.**

For the horizontal distribution of SLA RMSE, large values over 20 cm are mainly located in the area of

NEC pathway, continental shelf of the northeastern SCS and northeast of LUS, with maximum value of

32.7cm in v1(Fig.11b). In v1.3 (Fig.11c), large values in the area of NEC pathway almost disappeared,

the maximum RMSE is 30.3cm and most of values are less than 20cm, which may be interpreted as better

representing of NEC pattern due to movement of model's eastern lateral boundary. By comparing with

v1.3 or even v1, SLA RMSE decreases dramatically for the whole model domain and does not show





areas with obvious large values, and its maximum value is only 18.2cm, with most of values less than

10cm.

### 4.3 T/S profiles

For 3D T/S distribution, by comparing model results with climatology T/S profiles, the results from first

three versions show poor correlation with observations (Fig.12a and Fig.13a) and large RMSE (Fig.12b

and Fig.13b), i.e. 1.44-1.75°C for T and 0.13-0.14 for S (Table 2), even if they decrease with model

updates. Especially, for the vertical distribution, the RMSE can reach to larger than 3°C for T and 0.3 for

S in thermocline and halocline, respectively, and remained larger than 1°C for T in deep layer and 0.1

for S in above 700m depth (Fig.12d and Fig.13d). This may result from spurious diapycnal mixing due

to UCI schemes combination employed. Those updates in v1.1 and v1.2 can only slightly improve 3D

T/S, and cannot contribute to their intrinsic improvement, neither for surface forcing nor for lateral

boundary conditions, except for surface layer of shallower than 100m.

However, once AAG schemes combination employed in v1.3, the improvements to 3D T/S are obvious

with respect to the first three versions (Fig.12a,b and Fig.13a,b). The AC increases to 0.38-0.48 for T

and 0.30-0.44 for S, and RMSE decreases to 0.96-1.03°C for T and 0.10-0.11 for S, respectively (Table

2). For the vertical distribution, the AC remains around 0.4 for both T and S in the whole water column,

and over 0.6 for T in the surface layer (Fig.12c and Fig.13c), RMSEs significantly decrease to less than

2°C for T in thermocline and 0.25 for S in halocline, and less than 1°C for T and 0.1 for S in deep layer

(Fig.12d and Fig.13d).

For the horizontal distribution of 3D T/S RMSE, RMSE of T is more likely being more than 1.5°C with

maximum and minimum values being 4.45°C and 0.49°C (Fig.12e), and RMSE of S is larger than 0.1,

with





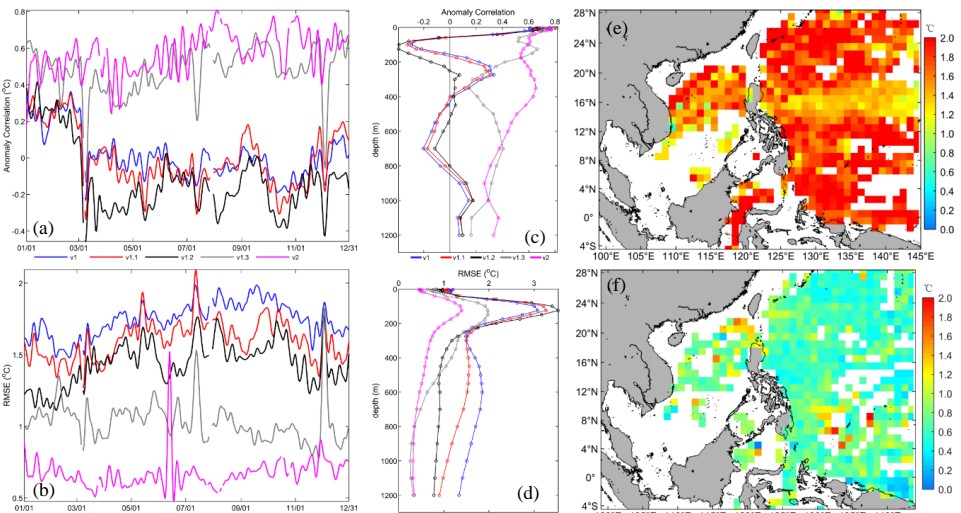

**Figure 12: (a) and (b) similar to Fig.10(a) and (b) but for T profile, respectively. (c) and (d) ) vertical**

**distribution of best estimates for each sub-version against observations as a function of depth, v1, v1.1, v1.2, v1.3 without data assimilation in 2013, and v2 with data assimilation in 2018. (e) and (f) similar to Fig.10(c) and (d), but for T profile in v1 and v2, respectively.**

maximum and minimum values being 0.81 and 0.06 (Fig.13e), in v1. Large values for S mainly locates in SCS and near equator in the Pacific Ocean. It is same with timeseries of RMSE, the horizontal

distribution of T/S RMSE show slightly decrease from version v1 to v1.2, but dramatic decrease in v1.3 (Figures not shown). Since benefiting from AAG schemes combination in v1.3, most of T RMSE is lower than 1.0°C, with maximum and minimum values being 1.72°C and 0.11°C, and most of S RMSE is less than 0.1 with maximum and minimum values being 0.62 and 0.03 in 2013, respectively.

By employing MOOAS, accuracy of 3D T/S has been improved continuously in v2 compare to v1.3 for

all the metrics in 2018 (Fig.12 and Fig.13). The mean AC has increased from 0.38 to 0.57 for T, and from 0.30 to 0.51 for S. The mean RMSE has decreased from 0.96°C to 0.67°C for T, and from 0.11 to 0.08 for S (Table 2). For vertical distribution of AC for T, it's over 0.6 in surface, over 0.4 above 600m, and over 0.3 in deep layer (Fig.12c). RMSE of T is less than 1.5°C for the whole vertical profile, and the maximum value is located at the thermocline similar with other versions, but the error decreases

dramatically (Fig.12d). Unlike T, vertical AC of S does not show significant improvement in v2 with respect to v1.3 below 200m, and it shows a little higher than which in v1.3(Fig.13c) in above 200m. S



RMSE is less than 0.25 for the whole vertical profile, with the maximum value located at surface and

decreasing with depth, and decrease to less than 0.05 below 600m (Fig.13d).

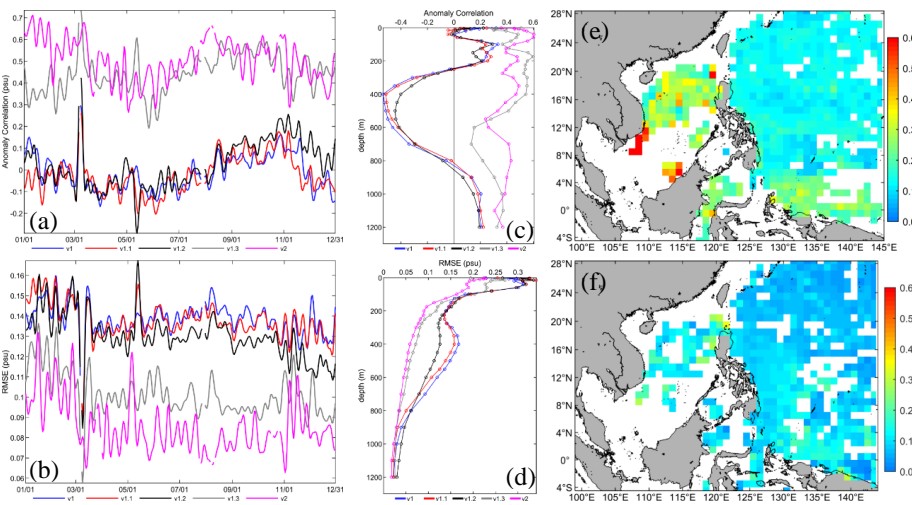

**Figure 13: Similar to Fig.12, but for S profile.**

For the horizontal RMSE distribution in v2, most of T RMSE is larger than 0.8°C with maximum and

minimum values being 1.96°C and 0.03°C (Fig.12f); and most of S RMSE is greater than 0.1, with

maximum and minimum values being 0.35 and 0.01 (Fig.13f), respectively, in 2018.

## 5.   Conclusions

This literature illustrates all the updates applied on SCSOFSv1 in both physical model settings, inputs

and EnOI data assimilation scheme in recent couple years based on the recommendations of Zhu et al.

(2016), such as land-water grid mask redistribution, data sources for bathymetry, initial condition, sea

surface forcing and open boundary condition changing to higher horizontal and temporal resolution,

moving the eastern lateral boundary westward, increasing vertical layers of model, and so on.

Three most sensitive updates are highlighted in this paper. Firstly, sea surface atmospheric forcing

method has been changed from direct forcing to BulkFormula to include effective negative feedback

mechanism of air-sea interaction by using COARE3.0 bulk algorithm. Upgrade leads to more reasonable



SST simulation with disappearing of abnormal values, significantly dropping of the maximum value of monthly mean differences between simulated SST and OSTIA, and decreasing of domain averaged

RMSE of monthly mean SST from 0.99-1.62°C in SCSOFSv1 to 0.87-1.15°C in BulkFormula run. The annual mean value decreases from 1.27°C to 1.00°C, indicating that the performance of model skill has improved by about 21%.

Secondly, tracers advection term discrete scheme UCI has been substituted with AAG in order to suppress spurious diapycnal mixing problem. After this substitution, the domain averaged monthly mean

temperature at 1000m layer decreases from 5.1°C to 4.5°C, and which of salinity decreases from 34.54 to 34.509, in January of the fifth model year, respectively. Even after 20 model years, domain averaged values of temperature and salinity increments are about 0.2°C and 0.03, suggesting that AAG schemes combination can well preserve the characteristics of water masses in deep ocean. In addition, model skill for SST also can benefit from AAG schemes combination with annual mean domain averaged RMSE

decreasing from 1.00°C to 0.77°C, showing 23% improving rate for the performance.

Thirdly, the original EnOI method in SCSOFSv1 has been upgraded to new MOOAS by adding four new functions. The multi-source observation data (SST, SLA, and Argo T/S profiles) can be assimilated simultaneously; Hanning high-pass filter is applied to ensemble members from 10 years free run while calculating the background error covariances to improve the dynamic dependency; FGAT method with

3-hour time slot is used to calculate the innovations; and IAU technology is employed with 7-day time window to apply analysis increment to the model integration in a gradual manner.

Moreover, scientific inter-comparison and accuracy assessment among five versions are conducted based on GOV IV-TT Class 4 metrics for four physical variables, SST, SLA, and T/S profiles. The improvement of accuracy of simulated SST mainly attributes to more accurate observed SST data source

used for sea surface heat flux correction, BulkFormula method for sea surface atmospheric forcing, AAG temperature advection discrete scheme. The improvement of SLA accuracy mainly benefits from good representations of NEC pattern caused by modification of model eastern lateral boundary, mean sea level air pressure correction, and T/S baroclinic structures improvement due to AAG employed. The


improvement of 3D T/S mainly benefits from AAG non-spurious diapycnal mixing schemes combination

employed.

At last, remarkable improvements for all above four variables are also benefited from MOOAS application. With respect to v1.3, domain averaged annual mean SST RMSE decreases from 0.66°C to 0.52°C with PI being 21.2%, SLA RMSE decreases from 12.9cm to 8.5cm with PI being 34.1%, T profile RMSE decreases from 0.96°C to 0.67°C with PI being 30.2%, S profile RMSE decreases from 0.11 to

0.08 with PI being 27.3%, in v2 while using MOOAS.

Although SCSOFSv2 has improved greatly comparing to the previous versions, some biases still exist in surface and subsurface. We need to continue to improve the system in both physical model settings and data assimilation scheme for next step, such as sub-grid parameterization scheme for unresolved physical processes, vertical turbulent mixing scheme to consider wave mixing, more accurate input and forcing

data source, and assimilating more or new types of observations (glider or mooring T/S, drifting buoys, *in-situ* velocity from moorings) into the system.

*Data availability.* No datasets were used in this article.

*Author Contributions.* XZ performed the physical model improvement and free-run simulations, designed and wrote the paper. XZ and ZZ updated MOOAS and performed the data assimilation

simulations. SR analysed and assessed model results. HW and YZ helped in reading and commenting on the paper. MZ helped in polishing the paper.

*Competing interests.* The authors declare that they have no conflict of interest.

*Acknowledgements.* This work was supported by "The Program on Marine Environmental Safety Guarantee" of "The National Key Research and Development Program of China" and the National

Natural Science Foundation of China under grant number 2016YFC1401409, 2016YFC1401605,





41806003. We would like to thank the anonymous reviewers for their careful reading of the manuscript
and for providing constructive comments to improve the manuscript.

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
