# Peer review of "The improvements to the regional South China Sea Operational Oceanography Forecasting System"

_Ocean Science, 2020_

## Referee Comment (RC1) · Anonymous Referee #1 · 14 Dec 2020

This article describes the progressive developments and improvements to a regional ocean forecast system of the South China Sea. The improvements demonstrated are worth reporting, and will likely be of interest to other researchers. But the level of detail and explanation is insufficient for the paper to really be valuable. With extra detail, the paper will likely be a good contribution to this field.

The main problem with this paper is that essential information is not included. Specific examples of exclusions follow.

Re: Ensemble Optimal Interpolation (EnOI) configuration

An important element of an EnOI system is the construction of the ensemble. This is

apparently described between lines 337 and 348. The details presented are unclear and don't sufficiently describe how the ensemble is constructed. After reading this passage a few times, I could speculate a few different ways the ensemble is constructed. This needs to be improved. How exactly were the ensemble members constructed? A clear way to describe this is like: "5-day averaged fields are subtracted from a 10year average", or "5-day averaged fields are subtracted from 60-day averaged fields", or similar. Then an interpretation could be given. For example, "the spatiotemporal scales represented in the ensemble of anomalies represent intraseasonal variability, or mesoscale processes".

The authors also don't describe the ensemble size, or whether covariance localisation is used. If localisation is used, what length-scales were chosen?

The authors don't say what observation errors are assumed – they merely mention in passing that estimates are made (L335).

The authors report improvements by assimilating more observation types (L332), constructing their ensemble differently (L337 – though, as noted above, the explanation provided is insufficient), introducing FGAT (L360), and by applying increments using IAU (L372). None of these techniques are new or novel. The statement: "Actually, it is close to impossible to calculate the synchronous innovations between the observation and model forecast entirely, since the temporal distributions of SLA and Argo data are irregular and variable at each analysis step", is untrue. Perhaps it's not a convenient calculation. But it is entirely possible, and implemented in many systems that use FGAT. In fact, the authors go on to explain how they did this (from L360). Re: Model configuration

One of the "solutions" implemented to address a problem with the model's mean circulation (see Figure 2) is to shift the eastern lateral boundary to the west by one degree. This excludes Guam Island from the configuration, and apparently results in an improved mean flow. This approach doesn't seem quite right. The authors have eliminated one element of the system (an Island), by making the model domain smaller. Exclusion of a real physical to improve the model's representation doesn't seem like a step forward. It would be better to understand how the presence of the Island influences the circulation, and then understand how the model can be reconfigured to more faithfully represent this influence.

Another change to the system is the adoption of bulk surface fluxes (see Figures 3 and 4, and section 3.1), rather than prescribed fluxes. This is a sensible change, but is not new or novel.

The authors refer to advection schemes, UCI and AAG. No reference is given, nor are these schemes described. Yet the authors identify the change from UCI to AAG as a key change that resulted in a substantial improvement in their system. The authors show that they significantly reduced the temperature and salinity bias in their system by adopting a different advection scheme (re: Figures 5, 6, and 7; and section 3.2). They refer to another study that showed the same result. This is again a good improvement, but again, it's not new or novel.

Re: results The new system seems to be better than all previous versions based on most metrics. Although this is not always the case. Figure 10 shows that for SST there is a period of 2-3 months when the new version has a smaller anomaly correlation compared to other versions (Figure 10). This needs to be explained.

re: data availability The authors claim that "No datasets were used in this article". This clearly isn't true. Perhaps the authors mis-understood what was expected here.

---

## Referee Comment (RC2) · Anonymous Referee #2 · 23 Dec 2020

Zhu, et al, present a look at changes to an existing operational South China Sea regional forecasting system, SCSOFSv1. They examine a number of changes to the system including: i) revision of the model grid and discretization; ii) change to the atmospheric forcing from direct forcing to utilizing the COARE bulk algorithm; iii) changing the advection scheme; and, finally, iv) making improvements to the ensemble optimal interpolation assimilation scheme.

The results show various improvements for the changes that have been made to the operational system. The manuscript is more of a technical report than a research article. While the results are not general, or improve our understanding of the science

of predictability in the region (dynamical analyses, etc.), there may be some useful information that would be valid to a specialized reader. So, in the interests of providing information, it is a paper that when in final form could be published in Ocean Science.

That said, the paper is fundamentally lacking in two primary ways. First, to be of any use to that select reader, it must provide details and explanation for the changes and how they would dynamically improve the representation of the region. Secondly, the paper lacks any rigorous analysis of the "improvements." Simply stating that one thing was changed and the summertime surface temperatures are cooler is not particularly useful.

First, I will go through the changes discussed by the authors as topics, then paper comments.

The first change made to the model is changing the model grid representation of the region. The bathymetry is is re-generated, smoothed, and the number of layers increase. The authors state that they smooth the bathymetry to reduce the bathymetric slope to an r-factor under 0.2. However, for an s-level model with various stretching function, the so-called "Haney" number is more valid and it increases with increasing the number of layers. There is no analysis as to whether this grid has improved spurious pressure-gradient flows or can better resolve the complicated bathymetric derived flows of the region. Figure 1 seems to show that the Luzon strait may be far too deep, which would alter the early Kuroshio flow with a more predominant intrusion through the strait. Secondly, the authors remove the island of Guam because it disrupted the flow in the v1 of the system. Guam lies as the southern island in the Mariana island arc that extends from the Mariana trench to the surface to impinge on the westward flow. Guam (as part of the archipelago) does disrupt the flow. If v1 of the system had an incorrect disruption, this implies that the boundary conditions used are not necessarily appropriate or account for the archipelago. More analysis and discussion of these items is needed to show that the new grid better represents the dynamics of the region.

The second change is the switch from direct forcing (momentum, heat, and salt) to utilizing the COARE bulk fluxes algorithm. Of course, the COARE algorithm has been in wide use for two decades in modeling, so the authors don't have anything new to present on the topic; however, the changes and differences seem to suggest not an improvement by the COARE algorithm, but rather a deficiency in the original direct forcing used by the original system. The COARE algorithm produces direct forcing using the atmospheric conditions and the SSS and SST of the model at the time of the forcing. This time dependency is obviously advantageous; however, to see such strong changes in the means shows that there was/is a forcing issue. The direct forcing should be nearly comparable, just lacking the ability to adjust to local intra-seasonal variance.

The third change was to switch advection schemes from a third-order, upstream in the horizontal and a central difference in the vertical to using the Akima scheme for both horizontal and vertical advection. The advection schemes, by design, represent the wavenumber spectrum differently, and in reading this manuscript, it is unclear why one would be preferred over another. Some cursory comparisons are provided, but how is this a result that would be useful to the community? There needs to be some explanation and analysis on how the flows are represented by each scheme. In the third-order scheme, it is far less sensitive to the prescription of explicit horizontal viscosity. What about in the Akima scheme? Did you vary your viscosity for momentum and diffusion for tracers? Is the sub-grid scheme chosen (authors did not mention which scheme, e.g., Mellor-Yamada 2.5, LMD, etc.) sensitive to the advection scheme?

The final change was to modify the ensemble optimal interpolation (EnOI). The authors reference an earlier paper, Zhu, et al., 2016, that does not contain details about their EnOI implementation. Oke, 2002, does a good job of explaining the EnOI methodology as a simplification of the ensemble Kalman Filter; however, it has a number of choices that must be made in its implementation. The authors of this paper do not describe any of their choices, their impacts, or reasoning. First, the choice of ensemble members. It is not well described in the manuscript. I will explain my understanding; however,

I am unsure how correct this is due to the lack of any detail. The authors break a 7 day assimilation window into 3 hourly periods. All observations are assigned to their nearest 3 hour slot, providing a total of 57 ensemble members. For each member, the innovations are calculated, and used for the EnOI setup. There is no discussion of localization, which is absolutely required in the ensemble scheme increments.

The authors take the increments and turn them into temporal nudging by dividing the EnOI derived increments by 57 and applying them weighted through time. The way it is presented would not seem to make sense. There are 57 innovations relative to the background. However, once an increment is made, the subsequent increments are now invalid because they are now being applied to a field with different innovations. This is why the original EnOI scheme applied the increments at the beginning of the window. Furthermore, localization is required to make this somewhat reasonable.

The final issue with this is that for a forecasting system, the authors never examine the forecasts. Adding forcing through time should reduce your forecasting skill because as soon as the forecast begins, the artificial forcing term disappears and the system is "shocked" back to its original state with the forcing that was "holding" it in place removed. How did these improvements change the forecast ability of the system? Does the EnOI system provide significant improvement over a model with the "best" atmospheric forcing?

\* Minor Comments

1. The authors are non-native English speakers, so I give some leeway, but the manuscript is difficult to read and confusing in many places that are too many to list here. Example lines: 68, 120, 123, 189, paragraph at 195, sentence at 210, 229, most of the EnOI description, 426,467, 473, 540, 545)

2. Figure 1: This map ratio is strange. There are roughly 25 deg in the vertical and 45 deg in the horizontal. The Philippine Sea is presented as smaller than the SCS. Likewise Figure 2 doesn't show the impact of changes on the SCS, the entire point of

the paper.

3. Paragraph near line 140: Why would you use a different set of initial conditions from the boundary conditions you are using? What is the persistence of the initial condition information? Over the 16 years or so that you spinup experiments, wouldn't the boundary conditions replace the initial conditions? There is no explanation for why you would use one product to initialize the model and SODA as your lateral boundaries.

4. Line 233 states that coastal sea surface forcing was "heating up the ocean". The region is massive with a deep ocean basin. How would some slight imbalances along a coast heat up the entire ocean of like 124 km2?

5. Line 241: Figure 4 is not a histogram. It is a bar chart.

6. Figure 5 caption, should this be SCSOFSv1 in (c)?

7. Paragraph around 275: You spin up the model until it is "in stable status". What does this mean? How do you determine "stable status"? During the spinup, you say that the temperature increases, but how does SODA compare? As per (3) above, are you not just converging towards the SODA state?

8. Line 360, for the EnOI, you generate seasonal background error covariances. Wouldn't it be more appropriate to have typical 7-day background error covariances? The covariance over a season is very large compared to the variability over 7-days. You should expect that your model is within a 7-day covariance period of the observations?

9. Section 4, "Scientific inter-comparison" I don't really see any scientific comparison, just comparing the bias, RMSE, and AC.

10. Table 2, what is the final row?

11. Line 460, what is "PI"?

12. Figure 11c, there are strong RMSE on either side of the Luzon Strait. The observations capture the surface bounce of the strong internal M2 tides there, resulting in a
25cm "error" that is present in altimetry because the surface bounce of internal tides vary in phase and location, meaning they aren't removed in the sea surface height measurements. So, 11c looks exactly as it should. But, 11d shows significant reduction in something that your model doesn't include. You don't have tidal forcing (at least explained in the manuscript), so you don't have the process present to reduce the error. This means that your EnOI is doing something non-physical to the system by trying to force an SSH expression that is due to internal tides.

––––––––––––––––––––––

---

## Editor Comment (EC1) · Andrew Moore (Editor) · 27 Dec 2020

Dear Author,

While the two referees note that there are some results in your paper that may be of interest to the community at large, there are some significant and substantial deficiencies in your manuscript that must be addressed before your paper can be reconsidered for publication or sent for further review. Both referees are of the opinion that important details are missing, and referee #2 further notes that your manuscript is lacking in "rigorous analysis."

[Figure]

Your paper will therefore require major revisions before the review process can continue.

Yours sincerely Andrew Moore

---

## Author Comment (AC1) · 18 Jan 2021

*This article describes the progressive developments and improvements to a regional ocean forecast system of the South China Sea. The improvements demonstrated are worth reporting, and will likely be of interest to other researchers. But the level of detail and explanation is insufficient for the paper to really be valuable. With extra detail, the paper will likely be a good contribution to this field.*

The authors thank the reviewer for the insightful comments, and we completely agree with all questions and comments raised by the reviewer, which have helped us to improve the quality of the manuscript, especially to achieve the level to be valuable publishing for the paper. We have tried to add more details and explanation of those improvements demonstrated in the revised manuscript.

*The main problem with this paper is that essential information is not included. Specific examples of exclusions follow.*

*Re: Ensemble Optimal Interpolation (EnOI) configuration*

*An important element of an EnOI system is the construction of the ensemble. This is apparently described between lines 337 and 348. The details presented are unclear and don't sufficiently describe how the ensemble is constructed. After reading this passage a few times, I could speculate a few different ways the ensemble is constructed. This needs to be improved. How exactly were the ensemble members constructed? A clear way to describe this is like: "5-day averaged fields are subtracted from a 10-year average", or "5-day averaged fields are subtracted from 60-day averaged fields", or similar. Then an interpretation could be given. For example, "the spatiotemporal scales represented in the ensemble of anomalies represent intraseasonal variability, or mesoscale processes".*

Thanks for pointing this out. The motivation of this paper is to introduce changes and improvements from SCSOFSv1 to SCSOFSv2. Many details about model configurations and EnOI systems were described in our previous papers (Zhu et al.(2016), Ji et al.(2015)). We have revised the whole paragraph to describe the construction of the ensemble in SCSOFSv1 and SCSOFSv2 in detail as reviewer's suggestions between lines 375 and 392 in the revised manuscript.

Modification:

L375-392: revised to "Secondly, we have introduced the method of computing the anomalies of ensemble numbers used for constructing the background error covariance following Lellouche et al. (2013). In SCSOFSv1, the anomalies are computed by subtracting a 10-year average from a long-term (typically 10 years) model free run snapshots with 5-day interval for the ocean state, i.e. sea surface height and three-dimensional temperature, salinity, zonal velocity, and meridional velocity. And the ensemble is selected within a 60 d window around

the target assimilation date from each year, adding up to about 130 members in total (Ji et al., 2015; Zhu et al., 2016). However, in SCSOFSv2, a Hanning low-pass filter is employed to create running mean according to Lellouche et al. (2013) in order to get intra-seasonal variability in the ocean state. Thus the anomalies are computed by subtracting the running mean with 20-day time window from a 10-year (2008-2017) free run daily averaged results. Especially, it is pointed out that the daily averaged free run results are selected within 60 d window around the target assimilation date from each year of 2008-2017 and used to compose ensemble members, thus about 590 members totally in SCSOFSv2. It means that the background error covariances rely on a fixed basis and intra-seasonally variable ensemble of anomalies, which improves the dynamic dependency."

*The authors also don't describe the ensemble size, or whether covariance localization is used. If localisation is used, what length-scales were chosen?*

Thanks for your reminding. We have revised the manuscript and pointed out that the ensemble size is 130 in SCSOFSv1 and 590 in SCSOFSv2. For the covariance localization, since it is mentioned in Ji et al. (2015), and has not been changed in SCSOFSv2, we did not mention in the original manuscript. We have clarified that the localization radius is 150 km in the revised manuscript between lines 409 and 410.

Modification:

L409-410: added "Meanwhile, the localization is still used with the radius set to be 150 km as in SCSOFSv1."

*The authors don't say what observation errors are assumed – they merely mention in passing that estimates are made (L335).*

Thanks for pointing out this issue. Please see the revised version between lines 371 and 374.

Modification:

L371-374: added "For the observation errors in SCSOFSv2, we simply set those of SLA and SST as constants of 0.09 cm and 0.5 ℃, respectively; as for those of Argo T/S, assuming they are represented as a function of water depth ($D$) following Xie and Zhu (2010) as $ERR_T(D)=0.05+0.45\exp(-0.002D)$, $ERR_S(D)=0.02+0.10\exp(-0.008D)$."

*The authors report improvements by assimilating more observation types (L332), constructing their ensemble differently (L337 – though, as noted above, the explanation provided is insufficient), introducing FGAT (L360), and by applying increments using IAU (L372). None of these techniques are new or novel. The statement: "Actually, it is close to impossible to calculate the synchronous innovations between the observation and model forecast entirely, since the temporal distributions of SLA and Argo data are irregular and variable at each analysis step", is untrue. Perhaps it's not a convenient calculation. But it is entirely possible, and implemented in many systems that use FGAT. In fact, the authors go on to explain how they did this (from L360).*

Thanks for the advice. The motivation of our paper is to demonstrate the technic details of a new forecasting system comparing with its precedent counterpart. We strongly agree with reviewer's point that none of these techniques are new or novel, but they are concurrently implemented for the first time in SCSOFSv2 with respect to its previous version SCSOFSv1. From our point of view, the primary objective of operational oceanography forecasting systems are more about accuracy than innovation, that's why we put our focus on the improvements of forecasting performances, rather than innovate techniques or methods only.

Although we did not develop new algorithm or parameterization, we consider it's a technical innovation by improving our operational forecasting system in the SCS in the way of implementing all those techniques in the new version of SCSOFS and increasing the forecasting accuracy.

For the statement about FGAT, we have revised the manuscript by adding more explanations between lines 404 and 410. We understand the fact that it is entirely possible to calculate by adding codes, but it would bring up the questions of significant increasing compute and storage cost. As for synoptic operational forecasting, we need to reach a balance between forecast accuracy and the computing and storage cost, so we consider 3-hour time slot used for calculating the innovations between the observation and model forecast with 1.5 hours misfit should be enough.

Modification:

L406: "running" revised to "run"; "the previous analysis" revised to " the previous analysis run".

L407: "minus" revised to "subtract".

L408: "time" revised to "temporal"

*Re: Model configuration*

*One of the "solutions" implemented to address a problem with the model's mean circulation (see Figure 2) is to shift the eastern lateral boundary to the west by one degree. This excludes Guam Island from the configuration, and apparently results in an improved mean flow. This approach doesn't seem quite right. The authors have eliminated one element of the system (an Island), by making the model domain smaller. Exclusion of a real physical to improve the model's representation doesn't seem like a step forward. It would be better to understand how the presence of the Island influences the circulation, and then understand how the model can be reconfigured to more faithfully represent this influence.*

Thanks for mentioning about this. We completely agree with the reviewer that our approach is simple and not quite right by excluding a real physical in science, instead of analyze how the model represent the island influence. As shown in Figure 2a, the NEC is split into two branches near the east lateral boundary due to the open boundary effects. For our focus domain is the interior of SCS, it's not that difficult to associate the solution of shifting the eastern boundary to the west to exclude Guam Island to get a reasonable large scale background circulation (NEC) for the interior of the SCS. Actually, we had done a serious of tests to decide optimal scheme of the eastern lateral boundary by comparing results of moving which westwards by 0.1°, 0.2°, 0.5°. We found that shifting the eastern lateral boundary westwards by 1° is the best option to get reasonable NEC pattern. This solution might seems simplistic, but is effective to improve the system. We could do more tests to understand how the presence of the Island influences the circulation, but it may beyond of the scope of this paper.

*Another change to the system is the adoption of bulk surface fluxes (see Figures 3 and 4, and section 3.1), rather than prescribed fluxes. This is a sensible change, but is not new or novel.*

Yes, we agree with the reviewer that the adoption of bulk surface fluxes is not a new or novel technique. But it is also an efficient approach to improve the forecasting accuracy in SCSOFSv2.

*The authors refer to advection schemes, UCI and AAG. No reference is given, nor are these schemes described. Yet the authors identify the change from UCI to AAG as a key change that resulted in a substantial improvement in their system. The authors show that they significantly*

*reduced the temperature and salinity bias in their system by adopting a different advection scheme (re: Figures 5, 6, and 7; and section 3.2). They refer to another study that showed the same result. This is again a good improvement, but again, it's not new or novel.*

In this paper, we use UCI to refer to the schemes combination of third-order upstream horizontal advection (U3H), fourth-order centered vertical advection (C4V) and horizontal mixing on epi-neutral surfaces for tracers(ISO), and AAG refer to the schemes combination of fourth-order Akima scheme for both horizontal and vertical advection (they are denoted as AA), and horizontal mixing on Geopotential surfaces (constant Z) for tracers (it is denoted as G), respectively. It has been described between lines 272 and 275 in the original manuscript. All schemes of U3H, C4V, Akima, mixing on epi-neutral surface, and mixing on geopotential surface, have been implemented in ROMS. We have given mode details about the differences between the Akima and four-order centered schemes between lines 309 and 313 in the revised manuscript. For most of ROMS community, the UCI schemes combination is the default and commonly settings. So the AAG schemes combination is proposed in this paper. We have finished many tests on various model settings to fix the the temperature and salinity bias, just put the optimal settings with the best results into this paper. And we are preparing a separate paper to show details of how tested model settings effect on this problem in order to keep the focus on the improvements of SCSOFS in this paper.

As we mentioned in previous replies, the motivation of our paper is to demonstrate the technic details of a new forecasting system compared with its precedent counterpart. The techniques and parameterization scheme are not new, but they are concurrently implemented by our forecasting system and perform better than previous version. In recent years, we found that many operational oceanographers mainly focus on how to improve the data assimilation technology when they want to improve the forecasting skills of the forecasting system, but pay less attention on the performance of physical model itself. Some forecasting systems are overly reliant on data assimilation while their pure physical models cant simulate reasonable large-scale circulation even. However, we think differently. Data assimilation should be adopted as a last resort under the premise that physical models cannot improve their forecasting skills. For example, the forecasting skill of sea surface temperature can be greatly improved by changing the surface forcing mode from prescribed fluxes to bulk surface fluxes (although it is not a very new or novel technique) as described in this paper, instead of increasing the amount of calculation as the introduction of data assimilation. Therefore, another main purpose of this paper is to tell the reader (operational oceanographer) that it will be once and for all to improve the forecasting skills of the system by improving the configuration of physical models, thereby to improve the prediction performance of the system in essence.

Modification:

L309-313: added "The fourth-order Akima scheme is a little different from the fourth-order centered scheme by replacing the simple mid-point average with harmonic averaging in the calculation of curvature term. Since the time stepping is done independently from spatial discretization in ROMS, the Akima scheme represents its advantage of reducing spurious oscillations, which arises with nonsmoothed advected fields, with respect to the fourth-order centered (Shchepetkin and McWilliams, 2003, 2005)."

*Re: results*

*The new system seems to be better than all previous versions based on most metrics. Although this is not always the case. Figure 10 shows that for SST there is a period of 2-3 months when the new version has a smaller anomaly correlation compared to other versions (Figure 10). This needs to be explained.*

Thanks for pointing this out. Actually, we have illustrated that v1, v1.1, v1.2, v1.3 are without data assimilation in 2013, but v2 is with data assimilation in 2018, in Figure 10's illustration. In order to simplify, we do not show the result from v1.3 without data assimilation in 2018. Now figureR1 shows those results from v1.3 (freerun) and v2 (with EnOI data assimilation) in 2018. It is found that the new system is still better than previous version based on anomaly correlation of SST in 2018.

[Figure]

FigureR1: The anomaly correlation of SST best estimates for v1.3 and v2 in 2018.

*Re: data availability*

*The authors claim that "No datasets were used in this article". This clearly isn't true. Perhaps the authors mis-understood what was expected here.*

Thanks for pointing it out. We have added it in the revised manuscript and removed some links in the text.

Modification:

L642-652: revised to "*Data availability.* GEBCO_2014 Grid, https://www.bodc.ac.uk/data/open_download/gebco/GEBCO_30_SEC/zip/, last access 3 January 2021; SODA 3.3.1, https://www2.atmos.umd.edu/~ocean/index_files/soda3.3.1_mn_download.htm, last access 3 January 2021; SODA3.3.2, https://dsrs.atmos.umd.edu/DATA/s oda3.3.2/REGRIDED/ocean/, last access 3 January 2021; CFSR, http://rda.ucar.edu/datasets/ds093.0, last access 3 January 2021; CFSv2, http://rda.ucar.edu/datasets/ds094.0, last access 3 January 2021; NCEP_Reanalysis 2, https://www.psl.noaa.gov/data/gridded/data.ncep.reanalysis2.html, last access 3 January 2021; AVHRR, http://www.ncei.noaa.gov/data/sea-surface-temperature-optimuminterpolation /v2.1/access/avhrr/, last access 3 January 2021; OSTIA, SST of *in-situ* drifting BUOY, AVISO along-track SLA, and Argo temperature and salinity profiles, https://marine.copernicus.eu/, last access 3 January 2021.

**Reference:**

Beckmann, A., Haidvogel, D.B.: Numerical simulation of flow around a tall isolated seamount. Part I: problem formulation and model accuracy. J. Phys. Oceanogr. 23, 1737-1753, 1993.

Ji, Q., Zhu, X., Wang, H., Liu, G., Gao, S., Ji, X., and Xu, Q.: Assimilating operational SST and sea ice analysis data into an operational circulation model for the coastal seas of China. Acta Oceanol. Sin., 34, 54-64, 10.1007/s13131-015-0691-y, 2015.

Li Ang,Zhang Miaoyin,Zhu Xueming*, et al. A research on the optimal approach of CFSR surface flux data correction based on different surface forcing modes. Haiyang Xuebao,2019, 41(11):51–63,doi:10.3969/j.issn.0253–4193.2019.11.006. (In Chinese with English abstract)

Marchesiello, P., Debreu, L., and Couvelard, X.: Spurious diapycnal mixing in terrain-following coordinate models: The problem and a solution, Ocean Model., 26, 156-169, 10.1016/j.ocemod.2008.09.004, 2009.

Shchepetkin, A. F., and McWilliams, J. C.: A method for computing horizontal pressure-gradient force in an oceanic model with a nonaligned vertical coordinate, J. Geophys. Res., 108, 3090, 10.1029/2001JC001047, 2003.

Shchepetkin, A. F., and McWilliams, J. C.: The regional oceanic modeling system (ROMS): a split-explicit, free-surface, topography-following-coordinate oceanic model, Ocean Model., 9, 347-404, 10.1016/j.ocemod.2004.08.002, 2005.

Xie, J., Zhu, J.: Ensemble optimal interpolation schemes for assimilating Argo profiles into a hybrid coordinate ocean model, Ocean Modelling, 33(3 − 4): 283 − 298, 10.1016/j.ocemod.2010.03.002.

Zhu, X., Wang, H., Liu, G., Régnier, C., Kuang, X., DakuiWang, Ren, S., Jing, Z., and Drévillon, M.: Comparison and validation of global and regional ocean forecasting systems for the South China Sea, Nat. Hazards Earth Syst. Sci., 16, 1639-1655, 10.5194/nhess-16-1639-2016, 2016.

---

## Author Comment (AC2) · 18 Jan 2021

**Reply on Anonymous Referee #2**

Zhu, et al, present a look at changes to an existing operational South China Sea regional forecasting system, SCSOFSv1. They examine a number of changes to the system including: i) revision of the model grid and discretization; ii) change to the atmospheric forcing from direct forcing to utilizing the COARE bulk algorithm; iii) changing the advection scheme; and, finally, iv) making improvements to the ensemble optimal interpolation assimilation scheme.

The results show various improvements for the changes that have been made to the operational system. The manuscript is more of a technical report than a research article. While the results are not general, or improve our understanding of the science of predictability in the region (dynamical analyses, etc.), there may be some useful information that would be valid to a specialized reader. So, in the interests of providing information, it is a paper that when in final form could be published in Ocean Science.

The authors thank the reviewer's confirmation to our manuscript. We completely agree with the reviewer that this manuscript is more of a technical report than a research article. But we think that most of these changes to the SCSOFS are scientific based technology and can be supported by basic physical oceanography theories or numerical simulation technic. It would provide useful information and references to numerical modelers, the communities of ROMS or operational oceanography how to set their numerical models in later.

That said, the paper is fundamentally lacking in two primary ways. First, to be of any use to that select reader, it must provide details and explanation for the changes and how they would dynamically improve the representation of the region. Secondly, the paper lacks any rigorous analysis of the "improvements." Simply stating that one thing was changed and the summertime surface temperatures are cooler is not particularly useful.

Thanks for pointing out our manuscript's deficiencies. We have revised this manuscript by adding some significant and substantial information and details which was not given in the original manuscript, such as the difference between Akima scheme and fourth-order centered scheme, the configurations and many details of EnOI scheme. We also supplement some "rigorous analysis" lacked in the original manuscript, such as why the impact of moving eastern lateral boundary on the SCS, COARE 3.0 bulk algorithm does not work better than direct forcing at all time, the reason for results of Fig. 11d better than that of Fig.11c, and adding a new Figure 12.

**First, I will go through the changes discussed by the authors as topics, then paper comments.** The first change made to the model is changing the model grid representation of the region. The bathymetry is re-generated, smoothed, and the number of layers increase. The authors state that they smooth the bathymetry to reduce the bathymetric slope to an r-factor under 0.2. However, for an s-level model with various stretching function, the so-called "Haney" number is more valid and it increases with increasing the number of layers. There is no analysis as to whether this grid has improved spurious pressure-gradient flows or can better resolve the complicated bathymetric derived flows of the region. Figure 1 seems to show that the Luzon strait may be far too deep, which would alter the early Kuroshio flow with a more predominant intrusion through the strait.

We changed the model grid representation of the region to make the model fit the actual coastline better, since it is an operational forecasting system and will be used to provide real forecasting. We derived and re-generated bathymetry from GEBCO, which has higher horizontal resolution (30 arc-second) than ETOPO1 (1 minute) and should be more reasonable and closed to the real bathymetry according to the introduction in its website. We smooth the bathymetry to keep the slope parameter (r) under 0.2 by referencing Marchesiello et al. (2009) and Beckmann and Haidvogel (1993). ROMS has a new pressure-gradient scheme associated to a modified equation of state limits computational errors of the pressure-gradient, which can improve spurious pressure-gradient flows and better resolve the complicated bathymetric derived flows of the region, especially with the higher vertical resolution. In SCSOFSv2, we have increased the vertical layers from 36 to 50, and employed an improved double stretching function in order to preserve a sufficient resolution in the upper layer. The two grid stiffness ratios parameters, slope parameter (r) and Haney number, range from 0.22 and 9.78 in SCSOFSv1 to 0.17 and 13.80 in SCSOFSv2, respectively. We have added some statements in the revised manuscript between lines 132 and 139, and Lines 144-146. According to our experience, if the model run with significant errors of pressure-gradient, it would blow out very quickly, but our system has run more than 40 years stably. It indicates that our system is stable enough.

For the bathymetry of Luzon strait in Figure 1, we add a figure by zooming in the Luzon strait (FigR1). It shows that the maximum depth is about 3000-3500m in Luzon strait, it is normal. We also show a daily averaged surface velocity around Luzon strait on July 15, 2018 (FigR2). It does not show predominant intrusion through the strait.

FigR 1: Bathymetry around Luzon strait

---

## Author Comment (AC4) · 18 Jan 2021

I would like to add two more figures.
* * *
[Figure]

[Figure]

**Fig. 1.** Figure2 with enlarging the domain to include SCS

[Figure]

**Fig. 2.** Figure 12-Daily averaged SLA on January 15, 2018, from AVISO, SCSOFSv1.3, and SCSOFSv2, respectively

---

## Author Comment (AC3)

**Reply to Editor Andrew Moore**

*Dear Author,*
*While the two referees note that there are some results in your paper that may be of interest to the community at large, there are some significant and substantial deficiencies in your manuscript that must be addressed before your paper can be reconsidered for publication or sent for further review. Both referees are of the opinion that important details are missing, and referee #2 further notes that your manuscript is lacking in "rigorous analysis."*
*Your paper will therefore require major revisions before the review process can continue.*
*Yours sincerely*
*Andrew Moore*

Thanks for your decision to let us make major revisions with our manuscript. We agree with all comments on our manuscript from two anonymous referees. We have tried our best to answer and response to all the comments and questions from referees. We have revised this manuscript by adding some significant and substantial information and details which was not given in the original manuscript, such as the difference between Akima scheme and fourth-order centered scheme, the configurations and many details of EnOI scheme. We also supplement some "rigorous analysis" lacked in the original manuscript, such as why the impact of moving eastern lateral boundary on the SCS, COARE 3.0 bulk algorithm does not work better than direct forcing at all time, the reason for results of Fig. 11d better than that of Fig.11c, and adding a new Figure 12.

In addition, the manuscript has been revised and proofread thoroughly according to the referees' comments by the co-author Ms. Miaoyin Zhang again, who is a professional of similar academic background with proficient English written skills and wrote her master thesis in English and received her master degree in England. Some paragraphs are rewritten and restructured, some figures (Fig.1, Fig.2, Fig.3, Fig.5, Fig.6, Fig.10, Fig.11, Fig.12, Fig.13, Fig.14) are redrawn to eliminate the countries boundaries in the land. Details of modifications are list as follow:

L11: "built up" revised to "constructed"

L13: "presents comprehensive updates had been conducted to" revised to "presents recent comprehensive updates of".

L14: deleted "in recent years"

L15: "sea surface" revised to "including sea surface"

L20: "version SCSOFSv1" revised to "version known as SCSOFSv1"

L45: "one or two decades" revised to "decade or two"

L47: "until 2015 in total" revised to "in total till 2015"

L54: "18km" revised to "18 km"

L108: "but the version of ROMS" revised to "while whose version"

L118: "a wall" revised to "closed boundary"

L120: "in the south of" revised to "across the south of

L121: "opened" revised to "involved"

L131: "bathymetry in the area near the coast of China mainland" revised to "topographic data in the coastal areas along China minland,";

L150: "open" revised to "opened"

L156: "changed" revised to "replace"

L157: "PSD" revised to "PSL"; "the website" revised to "their website";

L158: "http://www.esrl.noaa.gov/psd/ (Kanamitsu et al., 2002), to" revised to "https://psl.noaa.gov/ (Kanamitsu et al., 2002), with".

L159: "http://rda.ucar.edu/datasets/ds093.0," deleted.

L160: "http://rda.ucar.edu/datasets/ds094.0," deleted.

L161: " of them" deleted.

L164: "Barnier et al. (1995)" revised to "Barnier et al.(1995)'s method"

L167: "a constant number" revised to " a constant number of"

L168: "Meanwhile, we also replace the merged satellite's infrared sensors and microwave sensor, and *in-situ* (buoy and ship) data global daily SST (MGDSST) obtained from the Office of Marine Prediction of the Japan Meteorological Agency (JMA), with the infrared Advanced Very High Resolution Radiometer (AVHRR) satellite data, which is an analysis constructed by combining observations from different platforms on a regular grid via optimum interpolation and provided by National Centers for Environmental Information (NCEI)." revised to "Meanwhile, we use the infrared Advanced Very High Resolution Radiometer (AVHRR) satellite data in SCSOFSv2, which is an analysis constructed by combining observations from different platforms on a regular grid via optimum interpolation and provided by National Centers for Environmental Information (NCEI), instead of using the merged satellite's infrared sensors and microwave sensor, and *in-situ* (buoy and ship) data global daily SST (MGDSST) obtained from the Office of Marine Prediction of the Japan Meteorological Agency (JMA) in SCSOFSv1."

L180: "the NEC is separated into two branches since the model's eastern lateral boundary in SCSOFSv1, its main branch located at about 9.5°N-13°N, the other branch located at 14.5°N-17°N (Fig. 2a), which is clearly not in line with the fact." revised to "the NEC separated into two branches in SCSOFSv1 affected by model eastern lateral boundary setting, its main branch located at about 9.5°N-13°N, the other branch located at 14.5°N-17°N (Fig. 2a), which is clearly not in line with the fact."

L183: "This is because the Guam Island (Fig. 2, red circle) which located about (13°26′N, 144°43′E) is included in SCSOFSv1 and whose location is too close to the eastern lateral boundary." revised to "The cause for above result is that the Guam Island (shown in red circle in Fig. 2, located about 13°26′N, 144°43′E) is included in SCSOFSv1, whose location is too close to the eastern lateral boundary."

L191: "The left panel" revised to "The left panel (a)";

L192: "the right panel" revised to "the right panel (b)"

L193: "moving westward" revised to " moving 1 deg westward"

L203: "The modification for the time step is due to the change of the discrete schemes, which

would be illustrated in Section 3. Before the operational run, a 26 years climatology run is conducted for spinning-up, and is followed with a hindcast run from 2005 to 2018 (Wang et al., 2012). The daily mean of model results is archived and used to validate in the following parts of this paper." revised to "The reason for modifying time step is related to the change of the discrete schemes, which will be illustrated in Section 3. A 26 years climatology run is conducted for spinning-up at first, and followed by a hindcast run from 2005 to 2018 (Wang et al., 2012). The daily mean of model results is archived and used for subsequent evaluation."

L223: "the first option" revised to "one"

L225: "second" revised to "other"

L226: "the Fairall et al.(2003) COARE bulk algorithm" revised to "the COARE3.0 bulk algorithm (Fairall et al., 2003)"

L235: "would" revised to "could"; "one" revised to "method"

L236: "the second one" revised to "and the second"

L247: "Figure3 shows the distributions of monthly mean SST differences of SCSOFSv1, BulkFormula, SCSOFSv2 minus OSTIA SST in January, April, July and October, 2014 to stand for Winter, Spring, Summer and Autumn, respectively. It is found that the simulated SST are higher than OSTIA SST for all three results in general." revised to "Figure3 shows the distributions of monthly mean SST differences in January, April, July and October, 2014 to stand for Winter, Spring, Summer and Autumn, respectively. SST differences are calculated with SCSOFSv1, BulkFormula, and SCSOFSv2 subtracts OSTIA SST, respectively. It is found that the simulated SST are higher than OSTIA SST for all three sets of results."

L260: "bulk algorithm in both" revised to "bulk algorithm, which is employed in both"

L272: "algorithm, due" revised to "algorithm due"

L298: "(Fig.6b and 6c)" revised to "(Fig.6b and 6c) in deep layer"

L303: "on geopotential" revised to "along Geopotential"

L306: "of UCI and AAG based on" revised to "UCI in SCSOFSv1 and AAG in"

L308: "is in" revised to "reaches"

L351: "which is smaller" revised to "which is obviously smaller"

L352: "obviously" deleted

L353: " middle of the" revised to "central"

L357: "based on" revised to "judging from"

L429: "finish" revised to "finishes"; "output" revised to "outputs"L

L430: "is started" revised to "starts"

L431: "add" revised to "adds"

L458: "whole procedure" revised to "whole upgrading procedure"

L486: "SST for the whole" revised to "SST simulation for the whole"

L490: "It is noteworthy that" revised to "It is worth mentioning that"

L499: "smaller" revised to "less"

L536: "large values, and" revised to "large values in v2, and"

L543: Figure 12 added

L548: "Fig. 12a and Fig. 13a" revised to "Fig. 13a and Fig.14a"

L549: "Fig. 12b and Fig. 13b" revised to "Fig. 13b and Fig.14b"

L552: "in above" revised to "above"; "Fig. 12d and Fig. 13d" revised to "Fig. 13d and Fig.14d"

L557: "Fig. 12a,b and Fig.13 a,b" revised to "Fig.13 a,b and Fig. 14 a,b"

L560: "Fig. 12c and Fig. 13c" revised to "Fig. 13c and Fig. 14c"

L562: "Fig. 12d and Fig. 13d" revised to "Fig. 13d and Fig. 14d"

L564: "Fig. 12e" revised to "Fig. 13e"

L567: "Figure 12" revised to "Figure 13"

L571: "Fig. 13e" revised to "Fig. 14e"; "locate" revised to "locates"

L572: "in SCS" revised to "in the SCS"; "It is same" revised to "The trend is same"

L573: "show slightly" revised to "shows slight"

L574: "Since benefiting" revised to "Since it is benefited"

L579: "Fig. 12 and Fig. 13" revised to "Fig. 13 and Fig. 14"

L582: "Fig. 12c" revised to "Fig. 13c"

L584: "Fig. 12d" revised to "Fig.13d"

L585: "Fig. 13c" revised to "Fig. 14c"

L587: "Fig. 13d" revised to "Fig. 14d"

L589: "Figure 13: Similar to Fig.12" revised to "Figure 14: Similar to Fig.13"

L591: "Fig. 12f" revised to "Fig. 13f"

L592: "Fig. 13f" revised to "Fig. 14f"

L620: "technology" revised to "technique"

L637: "need" revised to "plan"

L671: "fi" revised to "fi"

L675: "Beckmann, A., Haidvogel, D.B.: Numerical simulation of flow around a tall isolated seamount. Part I: problem formulation and model accuracy. J. Phys. Oceanogr. 23, 1737-1753, 1993." added

L717: "Ji, Q., Zhu, X., Wang, H., Liu, G., Gao, S., Ji, X., and Xu, Q.: Assimilating operational SST and sea ice analysis data into an operational circulation model for the coastal seas of China. Acta Oceanol. Sin., 34, 54-64, 10.1007/s13131-015-0691-y, 2015." added

L742: "Li, A.,Zhang, M.,Zhu, X., Zu, Z., Wang, H.: A research on the optimal approach of CFSR surface flux data correction based on different surface forcing modes. Haiyang Xuebao,2019, 41(11):51–63,doi:10.3969/j.issn.0253–4193.2019.11.006 (In Chinese with English abstract)" added

L821: "Xie, J., Zhu, J.: Ensemble optimal interpolation schemes for assimilating Argo profiles into a hybrid coordinate ocean model, Ocean Modelling, 33(3 − 4): 283 − 298, 10.1016/j.ocemod.2010.03.002." added